# A CRISPR homing screen finds a chloroquine resistance transporter-like protein of the *Plasmodium* oocyst essential for mosquito transmission of malaria

Arjun Balakrishnan [1,2], Mirjam Hunziker[1,2], Puja Tiwary[1,2], Vikash Pandey[1,2], David Drew [3] & Oliver Billker [1,2,4] ✉

Genetic screens with barcoded *Plasmo*GEM vectors have identified thousands of *Plasmodium berghei* gene functions in haploid blood stages, gametocytes and liver stages. However, the formation of diploid cells by fertilisation has hindered similar research on the parasites' mosquito stages. In this study, we develop a scalable genetic system that uses barcoded gene targeting vectors equipped with a CRISPR-mediated homing mechanism to generate homozygous loss-of-function mutants after one parent introduces a modified allele into the zygote. To achieve this, we use vectors additionally expressing a target gene specific gRNA. When integrated into one of the parental alleles it directs Cas9 to the intact allele after fertilisation, leading to its disruption. This homing strategy is 90% effective at generating homozygous gene editing of a fluorescence-tagged reporter locus in the oocyst. A pilot screen identifies PBANKA_0916000 as a chloroquine resistance transporter-like protein (CRTL) essential for oocyst growth and sporogony, pointing to an unexpected importance for malaria transmission of the poorly understood digestive vacuole of the oocyst that contains hemozoin granules. Homing screens provide a method for the systematic discovery of malaria transmission genes whose first essential functions are after fertilisation in the bloodmeal, enabling their potential as targets for transmission-blocking interventions to be assessed.

Malaria is a vector-borne disease caused by *Plasmodium* parasites and transmitted by female *Anopheles* mosquitoes, primarily resulting in fatalities among children under the age of 5. Vector control strategies, such as the use of long-lasting insecticide-treated nets and indoor residual spraying have averted 68% of malaria cases between 2000 and 2015[1] but the emergence of insecticide-resistant mosquitoes and drug-resistant parasites have contributed to a reversal of this positive trend, resulting in a death toll of 608,000 as of 2022 (WHO Report 2022). These data underscore the necessity for novel strategies to combat malaria transmission which in turn requires a deeper understanding of the biological mechanisms that underpin malaria transmission.

In an infected host, haploid malaria parasites replicate asexually within erythrocytes. While haploid forms cause the disease, transmission is initiated by sexual precursor stages, the gametocytes, when a

[1]The Laboratory for Molecular Infection Medicine Sweden, Umeå University, Umeå, Sweden. [2]Department of Molecular Biology, Umeå University, Umeå, Sweden. [3]Department of Biochemistry and Biophysics, Science for Life Laboratory, Stockholm University, Stockholm, Sweden. [4]Umea Centre for Microbial Research, Umeå University, Umeå, Sweden. ✉e-mail: oliver.billker@umu.se

female *Anopheles* mosquito ingests an infectious bloodmeal. In response to environmental cues, gametocytes in the blood bolus mature rapidly into haploid gametes, which fertilise to generate a diploid zygote. The zygote replicates its genome and undergoes meiosis to produce a functionally tetraploid ookinete in which all four genomic products of meiosis persist within the same nuclear envelope. Motile ookinetes traverse the mosquito midgut epithelium and differentiate into oocysts. Over a period of 2 weeks in *P. berghei*, these cysts grow under the basal lamina and undergo endomitotic replication to generate and release infectious sporozoites, which invade the salivary glands of the mosquito. The significant population bottleneck posed by the transition from ookinetes to oocyst makes these stages vulnerable to transmission-blocking interventions, such as vaccines and anti-malarial[2–4]. Nonetheless, the oocyst remains one of the least understood life-cycle stages of *Plasmodium*[5].

Genetic screens using barcoded knockout vectors of the *Plasmo*GEM (*Plasmodium* Genetic Modification) resource have revealed gene essentiality at different haploid life-cycle stages of the rodent malaria parasite *P. berghei*[6–10]. *Plasmo*GEM mutants can be screened simultaneously in pools containing dozens to many hundreds of barcoded mutants, each disrupted in a different gene. Although barcoded pools have been transmitted through mosquitoes to identify developmental blocks at the subsequent liver stage, such experiments have not been effective at revealing oocyst gene functions[7] because all four products of meiosis persist and contribute to the proteome of the growing cyst. Once both parental genomes are active after fertilisation, loss-of-function mutations inherited from one parent are often functionally complemented by the intact allele inherited from the other parent[7]. Genomes only segregate days or weeks later when individual haploid sporozoites form within the oocyst. Most loss-of-function alleles will only reveal a phenotype days after that, depending on when any mRNA or protein that the sporozoite inherits from the cyst has turned over.

Here, we show that in the functionally diploid stages of *P. berghei*, gene functions can nevertheless be identified simultaneously and at scale by equipping barcoded knockout vectors with a guide RNA (*gRNA*)-based homing system. When integrated into one parental genome, a Cas9 endonuclease in the zygote can be targeted to the intact target locus of the other parental genome. In a pilot screen of 21 vectors, we demonstrate that gRNA homing is efficient enough to reveal new gene functions after fertilisation. We discover a structural homologue of the chloroquine resistance transporter (CRT), describe its importance for oocyst maturation and its localisation to a putative digestive vacuole compartment in the oocyst. This study unveils a new homing-*gRNA*-based genetic screening approach in *Plasmodium* that can shed light on post-zygotic gene functions and the biology of the oocyst, potentially identifying new targets for blocking parasite transmission.

## Results

### Single-sex *P. berghei* lines expressing fluorescence protein-tagged Cas9 are fertile

Sex in haploid malaria parasites is not chromosomally determined, and parasite clones, therefore, produce both male and female gametocytes. We reasoned that if we could genetically modify lines to produce gametes of only one or the other sex, these could be engineered further, such that in a genetic cross, each sex would separately deliver either a Cas9 endonuclease or a guide RNA to the zygote and cause a double-strand break in a target gene. Since *Plasmodium* parasites lack canonical non-homologous end joining[11], repair of double-strand breaks is always homology driven. We therefore further reasoned that a disrupted target allele not recognised by the guide and carried by one of the parental lines would serve as a repair template, leading to the disruption being copied from one parental genome to the other, generating a homozygous KO (Fig. 1).

To test this concept, we first created lines making only male or only female gametocytes by disrupting *female development 1* (*fd1*) or *male development 4* (*md4*), respectively (Supplementary Fig. 1), genes known to have sex-specific developmental functions early in gametocytogenesis[6]. These single-sex lines differed from those we recently used to screen for sex-specific fertility genes[10] in that they did not express GFP, and in that after negative selection against the resistance marker, an expression cassette for Cas9 fused to a blue fluorescent protein (BFP) was left behind in the disrupted *fd1* or *md4* locus under the control of an *hsp70* "constitutive" promoter (Supplementary Fig. 1A–C). Wild-type parasites were eliminated by fluorescence-activated cell sorting (FACS) on BFP, followed by limiting dilution. The resulting clones expressed a flag-epitope-tagged Cas9-BFP fusion protein of the expected size (Supplementary Fig. 1D), which was present in the nucleus of schizonts and, when lines were crossed, in zygotes and ookinetes (Supplementary Fig. 1E, F).

Clonal *md4⁻cas9-bfp* (female-only) parasites produced female gametes expressing the P28 surface marker (Supplementary Fig. 1F), did not produce male gametes (Supplementary Fig. 1G) and gave rise to ookinetes only when co-cultured with the *fd1⁻cas9-bfp* (male-only) line, which did release male gametes as expected (Supplementary Fig. 1G). These single-sex lines produced no oocysts when transmitted individually, but oocyst formation was restored when mosquitoes were fed on mice co-infected with both lines (Supplementary Fig. 1H), although not to the same level as wild type (Supplementary Fig. 1H). This was not due to crosses per se being less efficient at producing oocysts, because two similar single-sex lines not expressing Cas9-BFP[10] were fully fertile when crossed. This led us to ask if Cas9 expression is toxic, but a line expressing Cas9 from the same control regions but inserted into the *p230p* locus made large numbers of oocysts (Supplementary Fig. 1H). Further crosses between the different lines showed that a moderate reduction in oocyst numbers was associated specifically with the female-only *md4⁻cas9-bfp* clone, most likely due to a heritable variation in that clone that is unrelated to the expression of Cas9.

### Cas9-mediated genome editing after fertilisation is efficient

To quantitate the efficiency of Cas9-mediated genome editing after fertilisation at the level of individual oocysts, we designed a colour-swapping experiment. This involved constructing complementary single-sex lines in which the endogenous *myoA* gene was either fused to *mCherry* or to a *gfp*-encoding DNA sequence (Fig. 2a and Supplementary Fig. 2). MyoA is abundantly expressed during sporogony and tolerates c-terminal tagging[12]. We hypothesised that crossing single-sex lines expressing MyoA-mCherry and MyoA-GFP, respectively, should make yellow oocysts. However, if the latter locus additionally expressed one or more gRNAs targeting Cas9 specifically to the *mCherry* locus (Fig. 2a), homology-mediated repair of a double-strand break would use the complementary GFP-containing allele as a repair template, turning oocysts green.

Consistent with this concept, directional crosses between red and green parents produced yellow oocysts unless either the female or the male gamete provided gRNAs targeting the mCherry locus (Fig. 2b; all crosses also delivered Cas9 to the zygote). Since *md4* and *fd1* mutants do not undergo selfing, we were surprised to see 1–3% of oocysts in these crosses express only one parental allele, indicating that some zygotes may have obtained an incomplete genome from a parent, or that occasionally not all four products of meiosis survive or replicate in the oocyst. Our data suggest homing happens with 97% efficiency when the gRNAs are expressed from the female gamete, but this is reduced to 89% if the male carries the guides (Fig. 2b). This colour-swapping experiment demonstrates that we have developed a CRISPR method where the phenotype of the oocyst is dominated by only one parental allele, in this case *myoa-gfp*, irrespective of the route of inheritance.

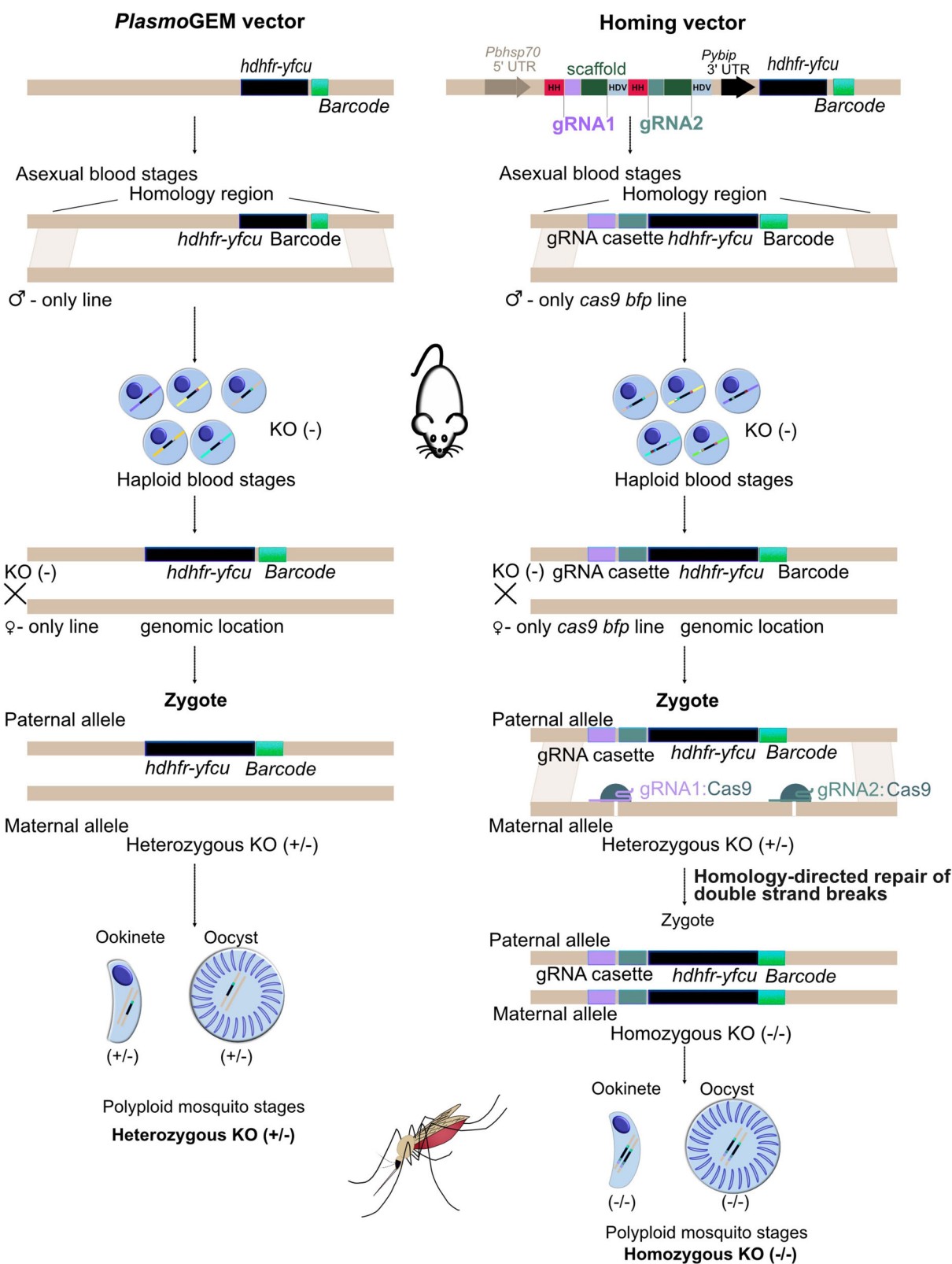

**Fig. 1 | Schematic representation of the homing strategy.** To generate homozygous knock-out ookinetes and oocysts (KO) after pooled transfection, modified *Plasmo*GEM vectors expressing two sgRNAs separated by hammerhead (HH) and hepatitis delta virus (HDV) ribozymes and Cas9 are brought together in the zygote at the point of fertilisation. gRNA1 and gRNA2 can function independently or in combination to induce a double-stranded break in the parental genome that has an intact allele. The disrupted allele in one parental genome is presumed to serve as a repair template for a Cas9-mediated double-strand break in the other parental genome.

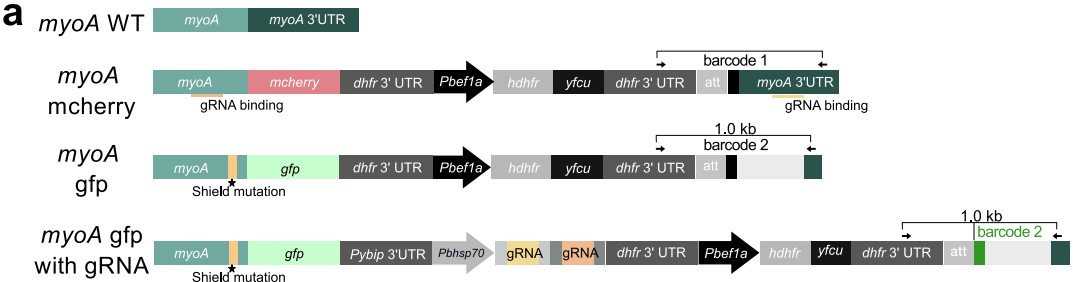

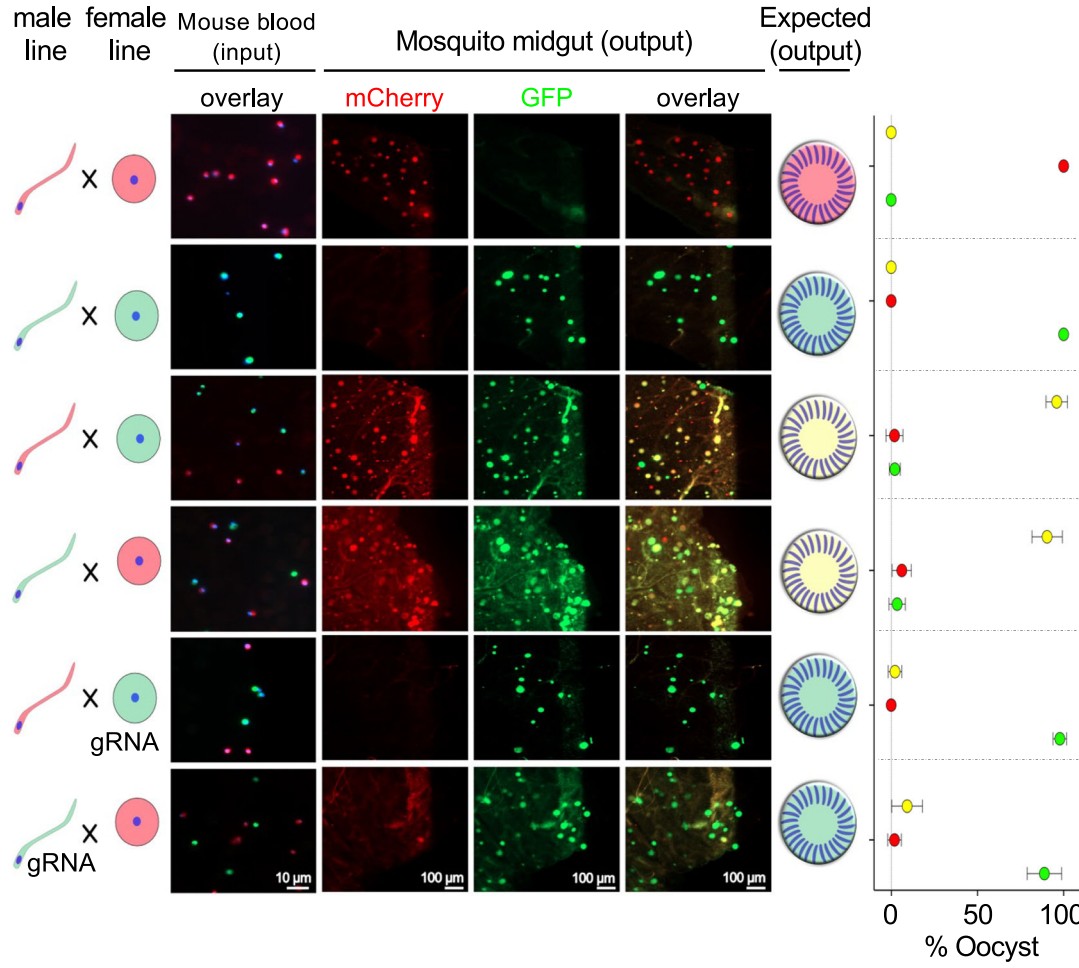

**Fig. 2 | Colour-swapping of tagged MyoA variants as proof-of-principle for homing postfertilisation. a** Schematics of myoA tagging vectors. The modified parasites carry an expression cassette for MyoA-GFP and two different gRNAs targeting the myoA gene and the myoA 3′ UTR, respectively, for improved cleavage efficiency. In tagged loci, the former target site carries a silent shield mutation, and the latter lacks the PAM site so that tagged loci can no longer be cleaved by Cas9. **b** Representative microscopic images of mouse blood (input) and oocysts on dissected mosquito midguts 12 days post-infection (output). Mouse blood was stained with Hoechst (blue) before imaging. Oocysts were scored as expressing GFP (green), mCherry (red) or both (yellow). Plots show the relative abundance of oocysts by colour as arithmetic means and standard deviations of 27–54 mosquitoes from two transmission experiments. To obtain accurate homing rates, each mosquito included in the analysis carried at least ten oocysts (mean 501 ± 180 standard deviation).

## Homing is scalable to screen for new post-fertilisation phenotypes

To ask if Cas9-mediated homing can reveal new loss-of-function phenotypes when pools of mutants are transmitted simultaneously, we inserted gRNA expression cassettes into each of 21 gene knockout vectors from the *PlasmoGEM* resource, using recombinase-mediated engineering[13,14] (Supplementary Fig. 3A). The resulting vectors used the CRISPR ribozyme-guide-ribozyme (RGR) strategy to express two *sgRNAs* from an RNA polymerase II promoter[15] and were additionally equipped with the gene-specific DNA barcodes used routinely in *Plasmo*GEM screens[16]. Target genes were selected to be redundant in the asexual blood stages[9], and to produce a range of known phenotypes to benchmark the system. At least three targets were included for each life-cycle stage. Additionally, eight less well-studied genes whose transcription patterns pointed at likely functions in the functionally diploid mosquito stages were examined[17]. Three well-characterised liver-stage essential genes known not to have a role in oocysts served as reference points against which stage-specific

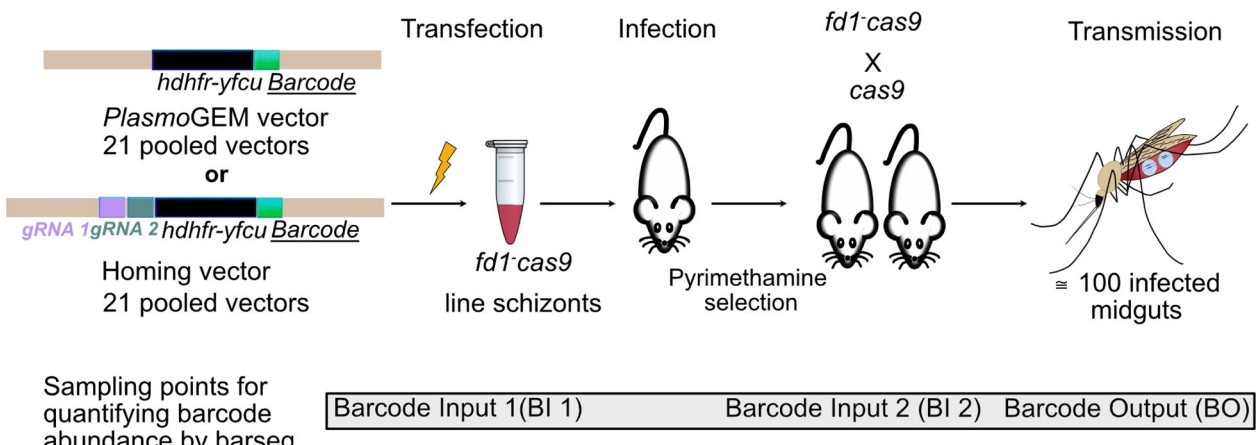

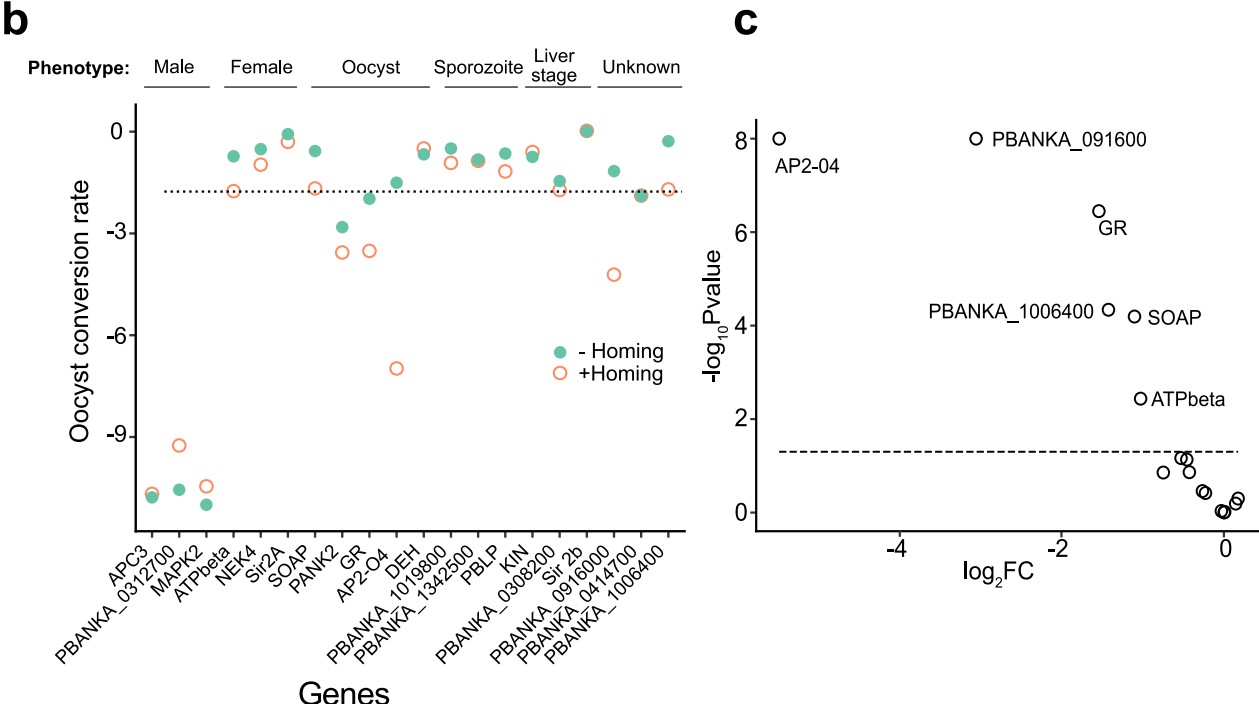

**Fig. 3 | Pilot screen using CRISPR homing. a** Schematic illustration of screen design. A male-only Cas9-expressing line was mutagenised with either homing or standard *Plasmo*GEM vectors and then crossed with a line providing female gametocytes (also expressing Cas9). Sampling points for barcode counting are indicated. **b** Oocyst conversion rates were calculated from the relative abundances of barcodes in the output (mosquito midgut, BO) vs. input (mouse blood, BI2) samples. Known stage-specific blocks of clonal mutants are displayed above the graph. **c** Volcano plot showing how homing affects barcode abundance at the oocyst stage, shown as $\log_2$ fold change in the presence of gRNA (+homing) compared to the no-gRNA control (-homing). Each circle represents a gene. Effect sizes are plotted against the $-\log_{10}$ $p$-values calculated using a $Z$-test. The dotted line shows a $p$-value of 0.05. Mutants with male fertility were lost from the pools irrespective of homing and were, therefore, omitted from this analysis. Data are from biological triplicates from a total of at least 100 mosquitoes per condition.

conversion rates of other mutants were calculated. Expression patterns of all selected genes are shown in Supplementary Fig. 4, known phenotypes and *P. falciparum* orthologues in Dataset 1.

We transfected *fd1⁻cas9-bfp* parasites either with a pool of the original *Plasmo*GEM vectors or with a pool of the corresponding homing vectors (Fig. 3a) and counted barcodes from the transfected vector pool (barcode input 1). Mutants selected with pyrimethamine (input 2) were then used to co-infect mice with a WT parasite line that constitutively expressed Cas9 under an *hsp70* promoter (*cas9*). This *cas9* line served as a donor for fertile female gametocytes instead of the less fertile *md4⁻cas9-bfp* used earlier. Female *Anopheles stephensi*

mosquitoes were then allowed to feed on six infected mice per pool. The cross between *fd1⁻cas9* and *cas9* parasites generated an average of 463 oocysts per mosquito midgut. For each of the two biological replicates, infected midguts of 80–100 mosquitoes per pool were dissected 12 days later. Barcodes counted in gDNA from the oocysts (barcode output) were compared to input 2, i.e., the barcode counts from the mice used to infect them (Fig. 3a). Around half of all oocysts will have arisen from selfing of the cas9 line, but these remained invisible to barcode counting.

To analyse the screen, we first asked if any *Plasmo*GEM homing vectors were underrepresented in the blood stages when compared to

their corresponding non-homing version (Supplementary Fig. 3B, C). While most homing and non-homing vectors were equally abundant at the point of transfection (Supplementary Fig. 3B), one homing vector targeting glycerol-3-phosphate dehydrogenase (G3PDH) was under-represented after drug selection of the transfected parasites (Supplementary Fig. 3C). Since G3PDH is dispensable in the blood stages, we concluded that the homing vector targeting this gene has either lost its functionality during recombineering or that one of the gRNAs has an off-target effect. This mutant was, therefore, removed from further analysis.

To find new gene functions in the oocyst, we looked for cases where the presence of gRNAs in the KO vector affected the conversion rate of asexual blood stages into oocysts (Fig. 3b). The oocyst conversion rate was first normalised to genes with liver-specific functions known to be dispensable at both blood and oocyst stage. Since a male-only line had been mutagenised, control mutants with known defects in male fertility, such as mitogen activated protein kinase 2 (reference [18]), were unsurprisingly depleted among the oocysts, irrespective of the homing capability of the vector (Fig. 3b). In contrast, for other genes homing vectors produced marked reductions in oocyst barcodes when compared with their non-homing equivalents (Fig. 3c). For the transcription factor *ap2-o4*, for instance, gRNAs caused a 64-fold reduction, consistent with its known essential role between ookinete formation and the oocyst[19]. Screen data for glutathione reductase (*gr*) and the secreted ookinete adhesive protein (*soap*) also reproduced known phenotypes of cloned mutants[20,21], which were not previously visible in *Plasmo*GEM screens.

For pantothenate kinase 2 (*pank2*), a complete loss of oocysts was expected[22], but the homing effect was smaller than for other known oocyst genes (1.7-fold) and did not reach statistical significance. This could mean that homing was not efficient enough in this mutant or that enough PANK2 protein or its enzymatic product was inherited from female gametocytes to overcome the essential oocyst function of this enzyme. The screen did not reveal a phenotype for 3-hydroxy acyl-CoA dehydratase (*deh*), possibly because the late oocyst degeneration observed in a cloned mutant in this gene[23] is not accompanied by a sufficient reduction in genome copies. Two of three oocyst-expressed genes of unknown function, PBANKA_0916000 and PBANKA_1006400, showed a marked homing effect, warranting further investigation. In summary, we conclude that the only gene for which the pilot screen is inconsistent with published knowledge is *pank2* and that both the drop-out rate and the false negative rate of the technology are acceptable for a future scale-up of the screening approach.

## PBANKA_0916000 encodes an essential digestive vacuole protein related to the chloroquine resistance transporter

How mutants behave in the homing screen may be confounded by factors other than gene function, such as the efficiency of gRNAs, how much transcript or protein a zygote inherits from the female parent and how fast these turn over. To validate the result of the pilot screen, we studied PBANKA_0916000, a poorly understood gene with a new phenotype, which we find encodes a chloroquine resistance transporter-like (CRTL) protein.

To understand the role of CRTL in mosquito stages, we first fused a c-terminal triple HA epitope tag to the endogenous gene (Supplementary Fig. 5A, B) to monitor the expression and localisation of the protein. Western blot analysis suggests absence of CRTL from asexual blood stages, but a full-length (132 kDa) fusion protein was detected in ookinetes (Supplementary Fig. 5C), which is consistent with mRNA abundance peaking in ookinetes according to the Malaria Cell Atlas[17] (Supplementary Fig. 5D). Fluorescence microscopy confirmed the absence of CRTL expression in both asexual and sexual stages, revealing a small number of weakly labelled foci that first appeared in the ookinete cytosol (Supplementary Fig. 5E). A time course of oocyst development detected CRTL-HA from day 2, where the label

colocalised with highly refractile crystals of malaria pigment (Fig. 4). Erythrocytic *Plasmodium* stages detoxify haem moieties from haemoglobin digestion[24] by aggregating them inside the digestive vacuole in crystalline form. In the gametocytes of *P. berghei*, malaria pigment forms in many small food vacuoles that distribute through the cytoplasm, while in *P. falciparum*, they remain perinuclear[25]. Pigment granules persist in 2-day oocysts, where they have been visualised by transmission electron microscopy[26]. In oocysts, pigment granules form typical patterns and are highly refractile when viewed by phase contrast or polarised light microscopy[25]. They are the oocyst's defining feature for microscopists to recognise even small cysts and distinguish them from mosquito midgut cells[27]. CRTL-HA colocalises with refractile pigment granules as they are spread throughout the cytoplasm on days 2 and 4 of oocyst development. From day 6, they form aggregates, which retain their original size while the oocyst grows. Around days 8–10, these labelled structures become forced into the narrowing spaces between growing sporogonic islands, where they continue to colocalise with the refractile pigment (Fig. 4), whose arrangement into lines or crosses is a hallmark of the oocyst at this stage. When oocysts sporulate, neither HA-tagged protein nor pigment granules get incorporated into sporozoites (not shown). These results are consistent with the tagged protein residing in the DV remnants surrounding the malaria pigment. To further confirm localisation of CRTL to the digestive vacuole (DV) within the oocyst, we examined the localisation of a known DV marker, parasitophorous vacuolar protein 1 (PV1), in the oocyst as previously described[28]. Unlike CRTL, PV1 is expressed in both blood stages and the oocyst (Supplementary Fig. 6A). In the blood stages, PV1 is present in the parasitophorous vacuole and the pigment-containing digestive vacuole of the schizont[28] (Supplementary Fig. 6B). Like CRTL, PV1 localises to the pigment-containing DV within the oocyst, providing additional confirmation of CRTL's DV localisation in the oocyst (Supplementary Fig. 6C).

To understand the function of CRTL better, we created a clonal *crtl* KO line in an mCherry expressing background (Supplementary Fig. 5A, F) and compared its growth to a GFP expressing WT (Pb Bergreen) clone. Although CRTL was first detected in ookinetes, mutant gametocytes converted to ookinetes in culture at the same rate as wild type (Supplementary Fig. 7B). Consistent with this, there was no difference in the number of oocysts between WT and mutant on day 10 or 14 (Fig. 5a), but mutant oocysts were smaller on both days (Supplementary Fig. 7A). In a time-course experiment, mutant oocysts stopped growing from day 8 (Fig. 5b, c). By day 18, WT oocysts had grown to more than twice the diameter of mutant oocysts (Fig. 5b). On the day when growth was first reduced, mutant oocysts had commenced DNA replication like wild type but were characterised by round vacuoles which excluded cytosolic mCherry protein (Fig. 5d). By day 10 virtually all mutant oocysts appeared vacuolated (Supplementary Fig. 7C). Time-lapse imaging showed Brownian movement of pigment crystals inside the vacuoles, suggesting vacuolation results from swelling of DVs (Supplementary Fig. 8, Supplementary Movies 1–4). In wild-type oocysts, pigment crystals colocalised with lysotracker, consistent with them residing inside an acidic compartment (Supplementary Fig. 8). In contrast, vacuoles of mutant oocysts did not stain with lysotracker, suggesting a swelling of the DV is accompanied by a loss of acidification. In contrast, ookinetes, which we showed not to require CRTL, possess lysotracker-positive acidic compartments also in the mutant (Supplementary Fig. 7D).

Transmission electron microscopy performed on 10-day-old oocysts confirmed that the smaller mutant parasites contained vacuoles of around 1 μm diameter (Fig. 5e). Unlike wild-type oocysts, where sporoblasts (Sb) and forming sporozoites (Sp) were visible on day 10, knock-out oocysts lacked signs of sporulation. Mutant parasites also failed to produce salivary gland sporozoites compared to wild type on day 20 (Fig. 5f). Mosquitoes infected with mutant were

 

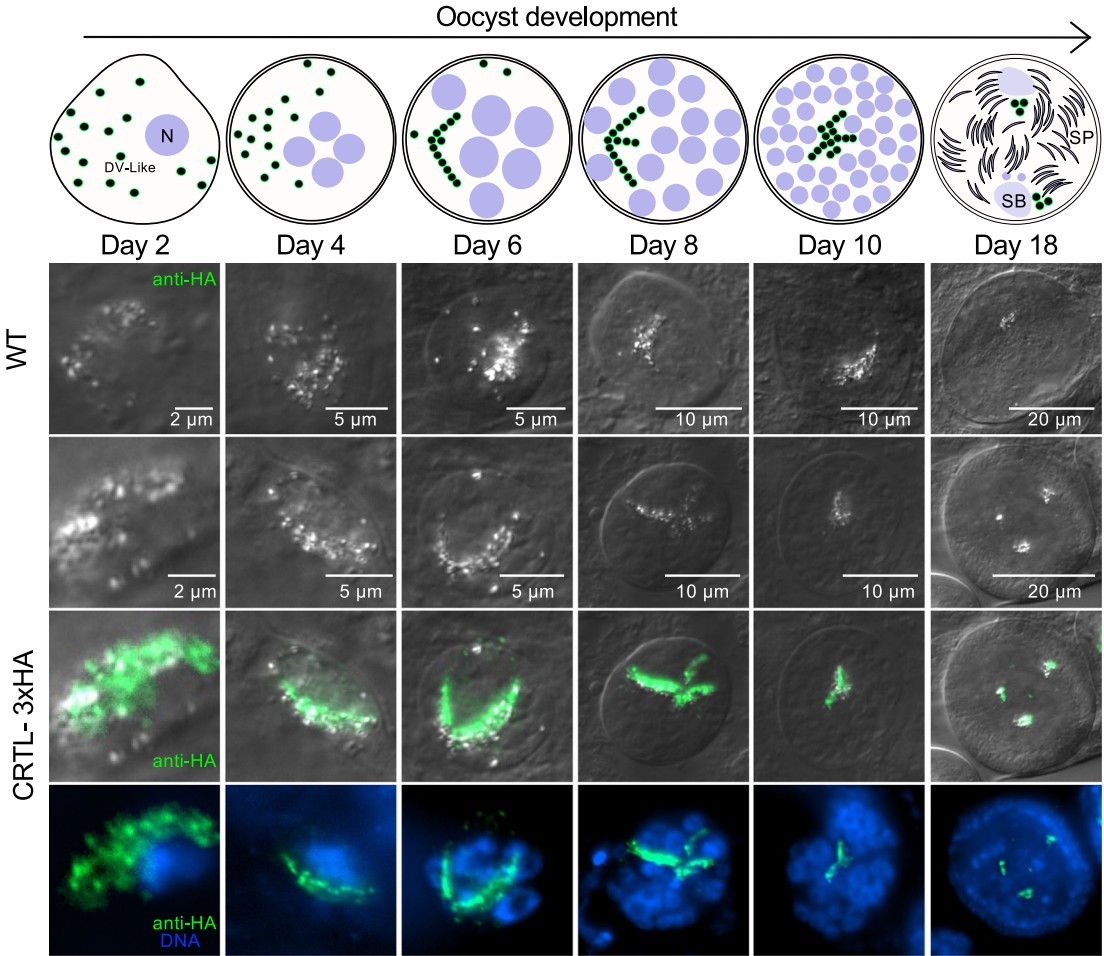

**Fig. 4 | Immunofluorescence micrographs localising CRTL-3xHA.** Midguts infected with wild-type parasites and parasites carrying a 3xHA tag on CRTL were dissected and fixed on different days post-infection. Immunostaining against HA is shown in green, and Hoechst-stained DNA is shown in blue. The schematic above illustrates the digestive vacuole-like structures (DV-like), nuclei (N), sporoblast areas (SB) and sporozoites (SP) at different stages of oocyst development. The data represent two separate transmission experiments, each involving the analysis of 25 infected mosquito midguts at the indicated time point.

unable to infect mice by bite, showing the gene is essential for transmission (Supplementary Fig. 7E).

An AlphaFold 2 model[29] predicts that PbCRTL has the drug-metabolite-transporter (DMT) fold[30], which in vertebrates is principally utilised by transporters for nucleotide-sugars and nucleotides[31] (Fig. 6a). In lower organisms, however, bacterial proteins (e.g., YddG, YdeD, and RhtA) have been shown to serve as exporters of amino acids and their derivatives[32–34].

Using FoldSeek[35], the closest experimental structure to PBANKA_0916000 is the Chloroquine Resistance Transporter from *Plasmodium falciparum* (*Pf*CRT)[36]. Indeed, a superimposition of the AF2 prediction and *Pf*CRT shows a high degree of structural similarity with a root mean square deviation of 2.9 Å[30] (Fig. 6b). The AF2-predicted structure of CRTL consists of ten transmembrane helices arranged in antiparallel pairs, similar to CRT. In addition to the core DMT-fold, the *Pb*CRTL protein has extensive loops and tails with intrinsically disordered domains (IDRs). Such extra-membranous regions are not required for transport, and it is most probable that CRTL forms a complex with other proteins that regulate its activity. The similarity between *Pb*CRTL and *Pf*CRT is mainly structural. *Pf*CRT (424 amino acids) aligns with the C-terminus of *Pb*CRTL (1113 amino acids), showing 18.1% identity and 32.7% similarity in their amino acid sequences. The N-terminus of *Pb*CRTL, which is rich in asparagine and lysine, does not form part of the predicted transporter domain. In contrast, the syntenic orthologs of *P. berghei* and *P. falciparum* are

much more alike, with 65.2% identity and similarity in the case of CRT and 34.7% identity and 47.1% similarity for CRTL. OrthoDB identifies CRTL homologues in all *Plasmodium* species but not in other Apicomplexa. An alignment of these homologues using ClustalW revealed high sequence similarity within the conserved core of the putative transporter (Fig. 6d, Supplementary Fig. 9).

## Discussion

The biology of *Plasmodium* ookinetes and oocysts and their interactions with the vector remain poorly understood, although these stages determine vectorial capacity and the reproductive rate of malaria. In this study, we demonstrate that a *gRNA*-based homing system makes functionally diploid mosquito stages tractable for the type of genetic screening approaches that have revealed thousands of gene functions in haploid stages[6–10,37]. In a proof-of-concept colour-swapping experiment, we found homing was 89% effective when the male gamete provided the gRNA and reached 97% when the gRNA entered the zygote with the female gamete. This difference may be due to the *hsp70* promoter that drives Cas9 expression being less active in males or to the compacted male nucleus delivering less Cas9-gRNA riboprotein complex to the zygote. Even the apparently less efficient design enabled a successful pilot screen when applied to a panel of 21 test genes since the screen revealed new gene functions in oocysts, overcoming a current roadblock to screening approaches.

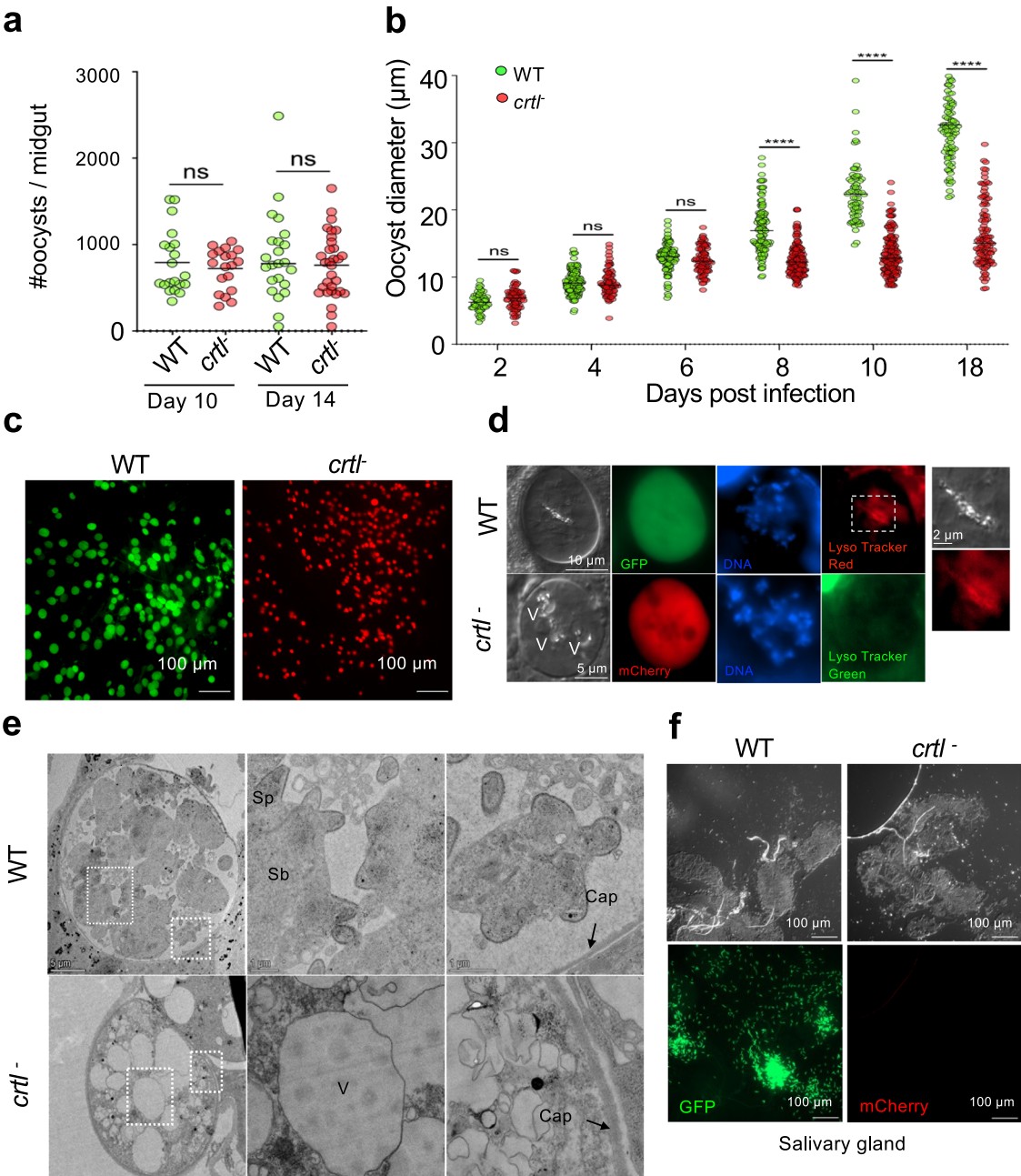

**Fig. 5 | Characterisation of a cloned CRTL mutant. a** Oocyst numbers on 25 dissected guts from infections with Pb Bergreen (WT, green) or *Pb mCherry crtl⁻* (red) parasites (*n* = 25). The data are representative of two transmission experiments. (ns not significant in unpaired *T*-test). **b** Growth kinetics of oocysts from 30 infected mosquitoes per time points. Oocyst diameter from randomly selected midguts was measured using ImageJ and plotted (*n* = 125 ± 50). (ns not significant; ***P* < 0.001; ****P* < 0.0001 in unpaired *T*-test) **c** Representative images illustrating oocyst size differences on a day 10 oocyst. **d** Representative images of day 8 oocysts highlighting acidic compartments with Lysotracker staining. The boxed area is enlarged on the right to show a DV-like acidic compartment in a WT oocyst. V vacuolar structures that appear in *crtl⁻* oocysts. **e** Representative transmission electron (TEM) micrographs of day 10 oocysts. Boxed areas are enlarged to show sporoblast (Sb), capsule (Cap) and sporozoite (Sp) in WT oocyst. *crtl⁻* oocysts are smaller, lack sporoblast and have enlarged vacuolar structures (V) despite having an intact capsule (cap). The data is from 2 independent replicates. **f** Representative images of a set of salivary glands of WT or *crtl⁻* infected mosquitoes 20 days post-infection.

For the type of screen we developed, genes must be targetable in one parent. Essential blood-stage genes, therefore, remain intractable for screens in mosquito stages, but genes that are essential for gametocyte differentiation or fertility in only one sex can be disrupted in the opposite sex and studied for functions after fertilisation. In the current pilot, we chose to mutagenise the male parent, allowing female fertility genes to still perform their functions at least up until fertilisation. A broader future screen of this type may therefore address systematically whether metabolic changes, such as the switch of mitochondrial metabolism to oxidative phosphorylation, are needed already in the ookinete or also later, in the oocyst.

Malaria parasites lost the canonical non-homologous end-joining machinery, and an alternative end-joining mechanism observed in *P. falciparum* is highly inefficient[11]. Off-target effects of gRNAs are therefore lethal already at the blood stage by causing irreparable chromosome damage, presumably at the ring stage, when no repair

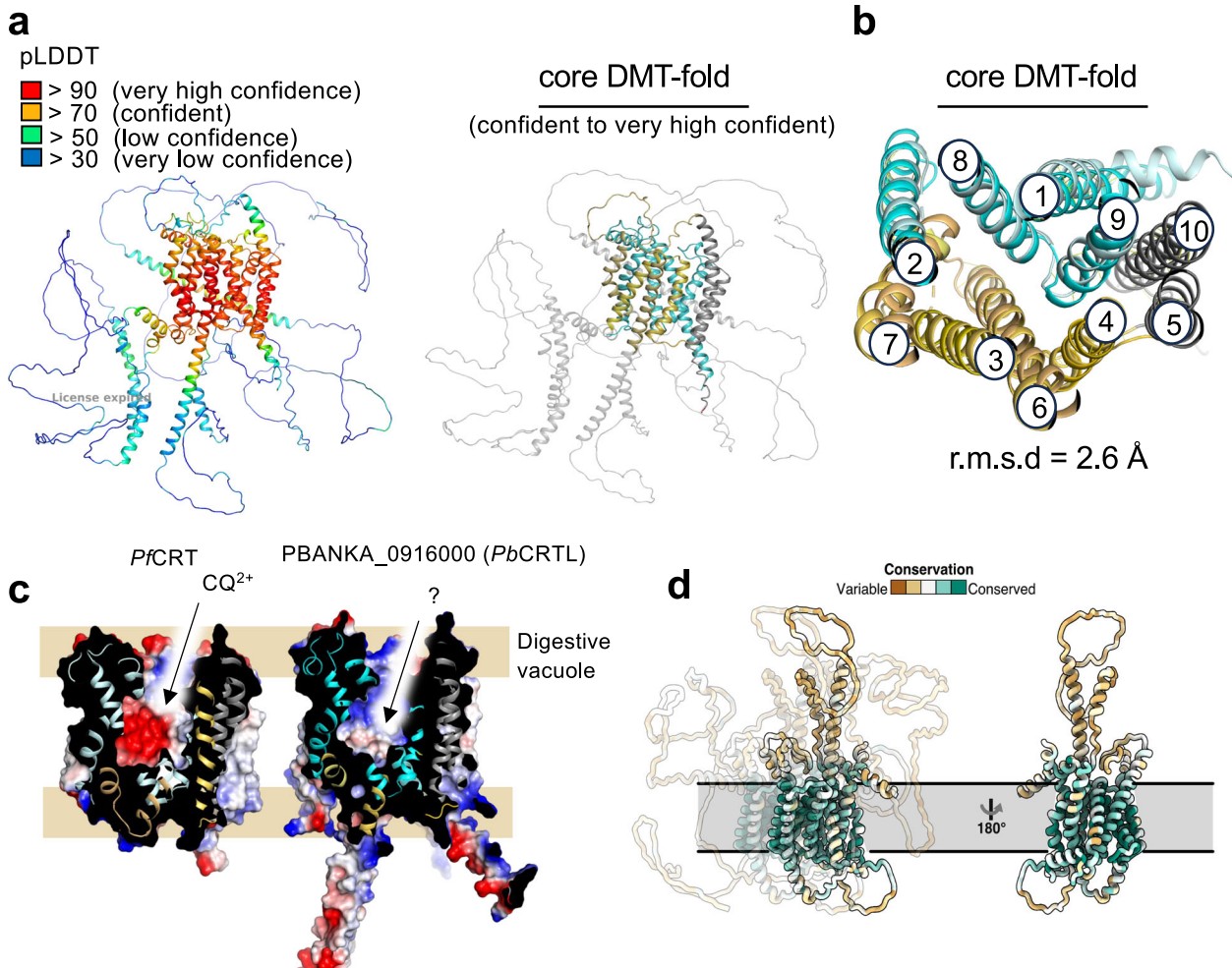

**Fig. 6 | CRTL structural modelling. a** Left: AlphaFold 2 (AF2) predicted structure of the PBANKA_0916000 encoded protein. Right: The core Drug-Metabolite-Transporter (DMT) region of the protein that is confidently predicted and coloured based on the two transporter bundle domains (orange and cyan) that alternate access using a rocker-switch mechanism. Regions of predicted disorder and extramembrane helices are shown in light grey. **b** Cartoon representation of a structural superimposition of *Pb*CRTL and *Pf*CRT (RMSD of 2.6 Å) viewed from the digestive vacuole lumen. The two bundles of the DMT-fold that alternate access are coloured in cyan and yellow for *Pb*CRTL and light-blue and sand for *Pf*CRT, respectively. TM5 and TM10 in grey can form an oligomeric interface in some members and show a larger variation. **c** An electrostatic potential surface representation of PfCRT and *Pb*CRTL protein highlighting the charge differences in the substrate binding cavity; coloured blue to red for positive to negative charge. **d** Two views of the core of *P. berghei* CRTL (as modelled on PfCRT) are shown and coloured according to conservation in the genus *Plasmodium* from brown (not conserved) to teal (high conservation). The conservation of each residue is based on the amino acid alignment in Fig. S7.

template in the form of other chromosome copies is available. Phenotypes in the homing screen are therefore probably not due to off-target effects but linked to the function of the target gene, i.e., we predict false positives to be relatively rare. This was borne out in the pilot screen, which did not produce phenotypes for genes known not to affect the development of midgut stages. In contrast, the absence of a phenotype may well be due to inefficient homing, which could be a property of the gRNAs used or possibly due to the chromatin state affecting the ability of Cas9 to access the target locus[38]. While most genes produced screening results consistent with their known functions, the true false negative rate of the screen design will require more data, which in future screens may come from looking at consistency across different members of a protein complex or enzymes of the same pathway and from corroborating negatives with additional gRNAs.

While it seems likely that CRISPR-Cas9 disrupts the second copy of its target between nuclear fusion and meiosis, i.e., within a few hours of fertilisation, we currently have no way of knowing the precise moment. Interpreting phenotypes in *Plasmodium* midgut stages is further complicated by maternal inheritance of mRNAs, proteins and organelles. As is typical in anisogamous species, the female developmental trajectory in malaria parasites is characterised by different types of posttranscriptional regulation. The zygote inherits from the macrogamete not only a metabolically reorganised mitochondrion but also translationally silenced mRNAs associated with ribonuclear particles[39,40]. Furthermore, the young zygote gives rise to a putative storage organelle, the crystalloid[41], the loss of which manifests only during sporogony[42]. Furthermore, a poorly understood pattern of paternal inheritance[43] has been reported. As with all conditional gene disruption experiments, a scaled-up homing screen will need to consider these complexities. Additional barcode readouts at the sporozoite stage will help assess the ability of mutant oocysts to complete sporogony and to detect any sporozoite phenotypes.

In the pilot screen, male fertility phenotypes from our earlier fertility screens[10] were reproduced independently of homing, which was expected since mutagenesis with a *Plasmo*GEM-gRNA vector was performed on the male parent. Where the zygote inherits a gene product (mRNA or protein) from the female gamete, our screen did

not reveal post-fertilisation functions. This was expected in the case of *nek4* because this protein kinase specifically regulates the zygotic cell cycle and is inherited from the female gamete[44]. The absence of a more marked homing effect for the beta subunit of the mitochondrial ATP synthase, another known female fertility gene[10], warrants further investigation. It raises the possibility that oxidative phosphorylation is more important in the early mosquito stages than in the growing oocyst, which would be relevant for efforts to target this pathway for transmission-blocking interventions[45,46].

The pilot screen revealed a role in oocyst development for a putative transporter, CRTL, which structural modelling showed to be a close homologue of *Pf*CRT. CRTL was investigated previously as a putative apicoplast protein but was not pursued further when found to be absent from asexual blood stages[47]. We now localise an endogenously tagged protein to the oocyst's pigment granules, where the most likely location of a protein with multiple transmembrane domains is in the DV, which would also be consistent with the localisation of CRT in blood stages. CRTL is clearly not part of the apicoplast or mitochondria, which have a different distribution in the sporulating oocyst[26,48] and get incorporated into sporozoites. Other DV proteins expressed in both blood and mosquito stages are multidrug resistance protein 2 (MDR2)[49] and PV1[28]. Here we show localisation of PV1 and CRTL to the DV within the oocyst. This confirms that CRTL is a unique oocyst DV marker.

Nutrient acquisition by the oocyst is poorly understood, and the role of its digestive vacuole needs to be explored further. Blood-stage parasites ingest haemoglobin from the host cell cytoplasm and degrade it proteolytically in the acidic environment of the DV. The prosthetic group, haem, is oxidised to ferriprotoporphyrin IX and detoxified by sequestration as microcrystals of hemozoin, a process which is targeted by CQ and other quinoline antimalarials[50,51]. Oocysts inherit the hemozoin crystals from the macrogametocyte, and it is tempting to speculate that a main function of their DVs may be to prevent the release of toxic haem from the hemozoin crystals contained within them. Lysotracker staining shows oocyst DVs are acidified, which would favour the self-association of haem[50]. Deacidified DVs in the CRTL mutant could thus result in the release of toxic haem, which may prevent further oocyst growth.

*Pf*CRT has been shown to export the doubly-charged chloroquine ion ($CQ^{2+}$) using the outwardly-directed $H^+$-gradient in the asexual blood-stage DV[36]. The available *Pf*CRT structure was captured with a cavity open to the digestive vacuole, revealing a highly negatively charged pocket for binding $CQ^{2+}$[36]. Although the natural substrate for *Pf*CRT has been proposed to include host-derived peptides of 4–11 residues[52], the uptake kinetics in *Xenopus* oocytes is slow and yet to be validated using purified components. Similar to bacterial homologues like YddG, it has been also shown that the purified *Pf*CRT reconstituted into liposomes can export positively-charged amino acids like arginine, with transport rates that seem more consistent with a physiological substrate[36]. Unlike *Pf*CRT, however, the cavity for *Pb*CRTL has both non-polar and positively-charged regions, indicating its substrate is rather a negatively charged metabolite[53], possibly with a hydrophobic extension, e.g., ATP (Fig. 6c).

How CRTL contributes to the acidification of the DV remains to be investigated and may well be related to its unknown substrate specificity. In *P. falciparum* asexual blood stages, the DV is acidified through a vacuolar $H^+$-ATPase and $H^+$-pyrophosphatase[54], generating the electrochemical gradient of protons that PfCRT has been suggested to use as a driving force for transport[55]. Whether similar mechanisms operate in oocyst has to our knowledge not been studied, but as with CRTL, deletion of the CRT ortholog of *Toxoplama gondii* leads to bloated lysosomes[56], and *Pf*CRT mutations that cause CQR also lead to altered DV volume regulation in blood stages[57]. While more work is needed to understand the function of CRTL, our data demonstrate how the scalable new screening strategy can lead to the systematic discovery of

essential malaria transmission genes whose first essential functions are after fertilisation, thus enabling their potential as targets for transmission-blocking interventions to be assessed.

## Methods

### Murine model
Female BALB/c mice were 6 weeks old, and female Wistar Han IGS rats were 150 g upon purchase from Charles River Europe. Mice were group-housed as four cage companions, and rats as two cage companions in individually ventilated cages with autoclaved chow, sterile drinking water, and sterile wood chip bedding and paper towels for nesting. Environmental conditions were maintained at 21 °C and 55% relative humidity under a 12/12 h light-dark cycle. The animals were acclimatised for 1 week in our animal facility before the experiments. All procedures were performed according to the guidelines and approved protocols from the Umeå Centre for Comparative Biology (UCCB) at Umeå University under Ethics Permit A13-2019 and approved by the Swedish Board of Agriculture.

### Mosquito model
*Anopheles stephensi* mosquitoes were reared and maintained at 28 °C with 80% humidity under a 12/12 h light-dark cycle. After being infected with P. berghei parasites, the mosquitoes were kept at 19 °C and 80% humidity. Throughout the study, all mosquitoes were provided with an 8% fructose solution (supplemented with PABA and Methyl-4-hydroxybenzoate) and were anaesthetised using CO2 gas or put on ice before dissection.

### Parasite model
*P. berghei* lines used in this study are the wild-type reference clone cl15cy1 of *P. berghei* ANKA, the Pb Bergreen line, which expresses GFP from a silent intergenic locus on chromosome 6 under the control of the *hsp70* promoter[58] and the Pb Bern mCherry line, which uses the same promoter to express mCherry from the p230p locus[59]. The PV1-mCherry line used in the study carries an mCherry-3xMyc tag at the 3' UTR of PBANKA_0919100[28].

### Generation of marker-free single-sex lines expressing BFP-tagged Cas9
Fluorescence and drug resistance marker-free single-sex lines were generated by disrupting PBANKA_145480 (*fd1*) or PBANKA_010240 (*md4*), respectively[6], using a CRISPR-RGR strategy (ribozyme-guide-ribozyme) with two *sgRNAs* to target the gene of interest[15] (Supplementary Fig. 1a) in the *P. berghei* clone cl15cy1. To construct a plasmid to express a CRISPR-Cas9 transgene, we modified the existing CRISPR-RGR strategy with two single guide RNAs (sgRNAs) to target the gene of interest[15]. The sgRNAs were designed using Benchling, and the sgRNA cassette (containing the gRNA along with RGR) was synthesised (Azenta Life Sciences). Subsequently, the sgRNA cassette was cloned into the cas9 plasmid (MH_046), which carries a *hdhfr* selection cassette to generate PL_HC_014 (gRNA plasmid targeting md4) and PL_HC015 (gRNA plasmid targeting *fd1*; Source data file). DNA fragments encoding an *hsp70* promoter, N-terminal flag-tagged *cas9*, *bfp* and *dhfr* 3'UTR were cloned into a plasmid and flanked with 500 bp sequences targeting either *fd1* or *md4* locus (Supplementary Fig. 1a).

To generate the repair template, *hsp70* promoter, *cas9, bfp* and *Pbdhfr* 3'UTR gene sequences were amplified from plasmids Pb_MH21 and R6K-GW-BFP (PL_HC_001), respectively, using Advantage 2 Polymerase Mix (TaKaRa). Individual fragments were assembled by stitch PCR, digested with *Kpn*1 and *Sac*11, and ligated into an intermediated vector (pMisc 017) to generate PL_HC013. The *cas9-bfp* fragment was sub-cloned into two pre-synthesised plasmids (Azenta Life Science) containing 500 bp sequences upstream and downstream of either *fd1* or *md4* gene locus to generate plasmid carrying repair template PL_HC_018 and PL_HC_017, respectively.

To generate transgenic single-sex lines expressing Cas9, *P. berghei* schizonts were co-transfected with 1 μg of the Cas9-sgRNA vector and linearised cas9-bfp repair template targeting either the *md4* or *fd1* gene locus. For each transfection, 1 μl of isolated schizonts was mixed with 7 μl of DNA and 18 μl of P3 Primary Cell 4D-Nucleofector solution from Lonza. This mixture was then added to a well within a 4D-Nucleofector 16-well strip from Lonza and electroporated using the FI115 programme on the Amaxa Nucleofector 4D. Transfected parasites were promptly injected intravenously into BALB/c mice and were subjected to selection with 0.07 mg/mL of pyrimethamine in drinking water starting from day one post-infection. Pyrimethamine selection was terminated on Day 5, and mice were administered 5-fluorocytosine (1 mg/mL, Sigma) via drinking water to eliminate the CRISPR plasmid carrying the *dhfr/yfcu* selection cassette (negative selection). The insertion of *cas9-bfp* in the *md4* and *fd1* gene loci was confirmed by PCR. Cas9-BFP expression in the nucleus was visualised by live imaging of infected cells in the Zeiss Axio imager 2 fluorescent microscope. The images were analysed using Fiji.

### FACS sorting and dilution cloning of single-sex lines

To generate clonal lines of Cas9-expressing *fd1⁻* and *md4⁻* parasites, infected mice were euthanised when parasitaemia reached 0.1%. Blood (10 μl each) was collected in 200 μl of CLB buffer (PBS, 20 mM HEPES, 20 mM Glucose, 4 mM sodium bicarbonate, 1 mM EDTA, 0.1% w/v bovine serum albumin, pH 7.25). Erythrocytes were then selected by gating on forward/side-light scatter, and BFP-expressing parasites were gated in the BV421A channel using the BD FACS Aria III. These BFP-positive parasites were sorted into cold CLB and intravenously injected into BALB/c mice. Each clonal line was re-genotyped to verify modifications in the *md4* and *fd1* gene loci and to ensure the absence of the *dhfr*-resistance cassette (*Cas9* plasmid).

### Generation and genetic crossing of myoA-tagged lines

MyoA tagging vectors were engineered in the *Plasmo*GEM *myo*A gDNA library clone pbG01_2365g05 using lambda Red-ET recombinase-mediated engineering in *E. coli*[14]. In short, two sets of recombineering oligos carrying *attR1*, *attR2*, and barcode were designed that have 50 bp homology to the 3′ UTR of *myoA* gene. The second set of recombineering oligos had a recognised section within the forward primer to incorporate a shield mutation in the final tagging vector. These primers were employed to amplify the *zeo-PheS* cassette from the *pR6K-attL-zeo-phes-attR2* plasmid. The resulting PCR products were then utilised for recombineering to generate the intermediate vectors PL_HC_020 and PL_HC_021. PL_HC_020 harbours the zeo-phes selection integrated into the C-terminus of the myoA in the library clone, while PL_HC_021 contains a shield mutation at the 3′ end of the myoA gene and a portion of the 3′UTR removed to protect the gene from cleavage by gRNA1 and 2, respectively. The clones were positively selected using zeocin and were sequenced. These intermediate vectors were further modified into final transfection vectors using a Gateway recombinase reaction, integrating the *dhfr*-resistance cassette into the vector. The mCherry tagging vector was produced by performing a gateway recombinase reaction between PL_HC_020 and the *Plasmo*GEM *pR6K_mcherry* gateway plasmid, generating PL-HC_022 (Supplementary Fig. 2B). Two variants of myoA-GFP tagging vectors were generated. The first one (PL-HC_028) was produced by a recombinase reaction between the *Plasmo*GEM pR6K gfp gateway plasmid and PL_HC_021. The second GFP tagging vector (PL_HC_026) was generated by a recombinase reaction between a modified *Plasmo*GEM vector carrying a gRNA cassette targeting the myoA gene and 3′UTR with PL_HC_026 (Supplementary Fig. 2A, B). These vectors were used to transfect *fd1⁻ cas9* and *md4⁻ cas9* parasites, following the procedures outlined earlier. Transfected parasites were selected on pyrimethamine and were dilution cloned. Each clonal line was genotyped to confirm modifications in the *myoA* gene loci. Genetic crosses between these lines were conducted by infecting mice with various combinations of *myoA*-tagged male and female-only parasites at a 1:1 ratio. Infected mice were anaesthetised and utilised for the direct feeding of *A. stephensi* mosquitoes. The raw data for oocyst counts can be found in the source data file.

### DNA extraction for genotyping

To extract the parasite from the host blood, heparinised whole blood was collected by cardiac puncture when the parasitaemia was approximately 3%. To remove blood plasma, 100 μl of whole blood was pelleted by centrifugation for 8 min at $450 \times g$. The cells were then resuspended in 1 ml of RBC lysis buffer (150 mM NH₄Cl, 10 mM KHCO₃, 1 mM EDTA) and incubated on ice for 15 min. After lysis, parasites were pelleted at $450 \times g$ for 8 min and washed 2 times with 1× PBS, or until complete removal of red pigment of blood. The parasite pellets were either stored at −80 °C, or DNA from these parasite pellets was isolated using Thermo Scientific GeneJET Genomic DNA Purification Kit according to manufacturer instructions, and genotyping was performed using advantage 2 polymerase mix (TaKaRa). The genotyping strategies are described in Supplementary Figs. 1, 2 and 5. The primers used for genotyping are mentioned in the source data file.

### Western blot assay

Parasite pellets (generated as described for DNA extraction) were resuspended in 10 μl of parasite lysis buffer (SDS 4%, 0.5% Triton X-100, 0.5× PBS). The lysate was combined with 4× Laemmli loading buffer supplemented with 20% β-mercaptoethanol. Samples were boiled at 95 °C for 5 min and loaded onto a 4–20% Bio-Rad TGX precast gel. Electrophoresis was performed at 90 V for 2 h. Proteins were transferred onto a PVDF membrane in the Turbo Transblot (Bio-Rad) using the High Molecular weight programme (10 min, 2.5 A constant, up to 25 V). The membrane was blocked with 2% BSA overnight and was probed with antibodies against goat anti-FLAG (1:1000, Cell Signalling Technologies, D6W5B, #14793), alpha Tubulin (1:10,000, mAb, mouse) or rabbit anti-HA (1:1000, Cell Signalling Technologies) followed by a compatible HRP-linked secondary antibody. The blots were developed using Immobilon Western Chemiluminescent HRP Substrate (Merck-Millipore) and were visualised on an Amersham Imager 680.

### Exflagellation and ookinete conversion assay

BALB/c mice were intraperitoneally injected with 200 μl of phenylhydrazine (6 mg/ml; Sigma). Three days after treatment, the mice were infected with *P. berghei* parasites. Blood from infected mice exhibiting 8–10% gametocytaemia was added to ookinete medium (20% FBS, 100 μM xanthurenic acid (Sigma), 24 mM sodium bicarbonate, and RPMI (Gibco 52400-025), pH 8.2). For the exflagellation assay, infected blood was added at a 1:5 ratio of blood to ookinete medium, and parasites were incubated for 10 min. Male exflagellation events were counted in a standard haemocytometer under a light microscope. To determine ookinete conversion rates, infected blood was added to ookinete media at a 1:5 ratio and incubated for 23 h at 19 °C to allow for ookinete formation. After incubation, 1 ml of ookinete culture was taken and mixed with 1 ml of 4% paraformaldehyde (PFA) and incubated at room temperature for 15 min. Fixed ookinetes were pelleted at $500 \times g$ for 3.5 min and washed twice with 1× PBS. The final pellet was resuspended in 100 μl of PBS. From the resuspension of the fixed ookinete, 15 μl was mixed with 50 μl of staining solution (Cy3-labelled 13.1 mouse monoclonal anti-P28 (1:500 dilution) and Hoechst (1:2000)). The samples were incubated at room temperature (RT) for approximately 12 min. 6 μl of the stained sample was placed on a glass slide and covered with a Vaseline-edged coverslip. The samples were visualised in a Zeiss Axio Imager 2 fluorescent microscope. The images were analysed using Fiji. The number of ookinetes (banana-shaped cells that are Cy3-positive) and female gametocytes (round cells that

are Cy3-positive) were quantified. The conversion rate was calculated as the percentage of Cy3-positive ookinetes to Cy3-positive macrogametes and ookinetes.

## Indirect Immunofluorescence

Blood-stage parasites and ookinetes were fixed in 4% PFA in PBS for 40 and 15 min, respectively, at RT. Fixed cells were washed with 1× PBS and resuspended in PBS. Cells were added to poly-D-lysine coated glass slides and were allowed to settle for 15 min. Permeabilisation of the parasites was done using 0.2% v/v Triton X-100 in PBS for 5 min at RT. Permeabilised parasites were washed with 1× PBS and then blocked with 3% w/v bovine serum albumin (BSA) in PBS for 40 min at RT. Parasites were stained overnight with 13.1 mice monoclonal anti-P28 (1:1000) and rabbit anti-HA (1:250, Cell Signaling Technologies) antibodies. Parasites were washed 3 times in PBS for 10 min each and then were stained with secondary antibodies (Thermo Fischer) anti-mouse Alexa Fluor 647 (1:500) and anti-rabbit Alexa Fluor 488 (1:500) for 2 h. The cells were washed and stained with Syto 19 or Hoechst to visualise nuclear DNA. Images were acquired using a Leica SP8 confocal microscope and were visualised using Fiji. For live cell imaging of blood and ookinete stage parasites, cells were stained with Lysotracker Deep Red (1 µM, Invitrogen) or Lysotracker Green DND-26 (1 µM, Invitrogen) for 10 min at RT and were counterstained with Hoechst to visualise DNA. 6 µl of the stained sample was placed on a glass slide and covered with a Vaseline-edged coverslip. The samples were visualised in a Zeiss Axio Imager 2 fluorescent microscope. The images were analysed using Fiji.

## *P. berghei* mosquito infection and transmission

Mosquitoes were fed on mice infected with *P. berghei* at a parasitaemia of 3–5%. To determine oocyst number and size, midguts were dissected on the indicated day post-infection and visualised using a Zeiss Axio Imager 2 fluorescent microscope. Oocyst numbers and size were quantified using built-in plugins in Fiji. Localisation of the HA-tagged protein in the oocyst was performed as previously described[60]. Briefly, infected mosquito midguts were fixed in 4% PFA containing 0.1% saponin in PBS and incubated for 45 min on ice. After three washes with PBS/0.1% saponin of 15 min each, parasites were blocked with 3% BSA, 0.1% saponin in PBS for 30 min on a thermal shaker at room temperature (RT). Parasites were stained overnight with rabbit anti-HA (1:200, Cell Signalling Technologies) and then stained with anti-rabbit Alexa Fluor 488 (1:500, Thermo Fisher Scientific) for 2 h. Cell nuclei were labelled with Hoechst. For live imaging of oocysts and visualisation of acidic compartments in the oocyst, the midgut was dissected into PBS containing Lysotracker Deep Red (1 µM, Invitrogen) or Lysotracker Green DND-26 (1 µM, Invitrogen) along with Hoechst and incubated for 10 min at RT before imaging with a Zeiss Axio Imager 2 fluorescent microscope. Time-lapse images of the oocysts were captured by imaging live oocysts over a period of 30 s in Zeiss Axio Imager 2. Salivary gland sporozoites in the infected mosquitoes were imaged on day 18 and day 20. To assess mosquito-to-mouse transmission, approximately 15–20 infected *A.stephensi* mosquitoes were fed on 3 anaesthetised BALB/c mice. Mouse parasitaemia was monitored until day 8 post-mosquito bite by Giemsa staining.

## Generation of homing vectors

Homing vectors were generated by recombineering the gRNA cassette to existing *Plasmo*GEM vectors. To do so, we generated a plasmid by Gibson assembly to generate an MCS (multiple cloning site) sandwiched between the *Pb hsp70* promoter and the *P. yoelii dhfr* terminator (PL_HC_030). The plasmid also had a zeocin resistance gene cloned adjacent to the *Pb hsp70* promoter. The ribozyme-guide ribozyme (RGR) sequences containing two sgRNA targeting the gene of interest were synthesised (Azenta Life Science) and were cloned into the MCS of PL_HC_030. The zeocin resistance gene, along with the

*hsp70* promoter, gRNA cassette, and *dhfr* terminator, was amplified with primers containing homology regions (50 bp each) to either side of the 3xHA part of the *Plasmo*GEM KO vector. The resulting PCR products were then utilised for recombineering into *Plasmo*GEM vectors to generate homing vectors.

## Generation of mutant pools using *Plasmo*GEM and homing vectors

To generate pools of *Plasmo*GEM and homing vectors, 21 transformed *E. coli* TSA cells carrying knock-out vectors were inoculated into 96 well plates containing 1 ml of Terrific Broth with kanamycin (30 µg/ml) and were grown overnight at 37 °C. The cultures were then pooled, and plasmid was extracted using the QIAGEN Plus Midi Prep Kit. A total of 30 µg (approximately 1 µg of each vector) was digested overnight with NotI to release the targeting vector from the linear plasmid backbone and was used for three independent transfections as described previously[9]. Briefly, *fd1⁻cas9* schizonts derived from infected rats were harvested after 22 h of culture and purified on a Nycodenz (Sigma) gradient. Purified schizonts were electroporated with either *Plasmo*GEM or a homing vector pool using the FI115 programme on the Amaxa Nucleofector 4D (Lonza). Transfected parasites were then injected intravenously into BALB/c mice and were selected with pyrimethamine (0.07 mg/mL) in drinking water from day one post-infection.

## Transmission of barcoded KO parasites through mosquitoes

After seven days post-transfection with the *Plasmo*GEM or homing vector pool, mice with a parasitaemia of 3–5% were euthanised, and blood was collected via cardiac puncture. The blood from *fd1⁻cas9* mutant parasites was mixed with a female donor line to establish a final ratio of infected red blood cells between these parasites at 1:1. This mixture was used to infect BALB/c mice. Mice with a parasitaemia of 3–5% on day 3–4 post-infection were selected to feed *A. stephensi* mosquitoes. For each biological replicate, 80–100 infected mosquito midguts were dissected for genomic DNA extraction. The female donor line used was *cas9*, which is capable of selfing but carries no barcode module.

## Library preparation, barcode sequencing, and analysis

DNA was extracted from three different samples. The vector pool DNA was extracted from the left-over media in the transfection cuvette (input 1). This served as input control and was used to quantify the initial diversity of the barcodes/vectors. Parasite genomic DNA was extracted from the mouse blood sampled after feeding the mosquitoes (input 2) using a DNeasy Blood and Tissue Kit (Qiagen). The parasite gDNA was extracted from the infected mosquito midguts 14 days post-infection (Barcode output) by boiling the midguts in Quick Extract DNA Extraction Solution (Epicentre). *Plasmo*GEM barcodes were amplified by PCR in using the extracted DNA samples and the primers 91_Illumina and 97_Illumina (Source data file). The PCR amplicons were used as an input for a second PCR to add sample-specific index tags using primers shown in the source data file. The resulting libraries were then pooled and sequenced using the Illumina MiSeq Reagent Kit v2 (300 cycles), which were loaded at a low cluster density ($4 \times 10^5$ clusters/mm2) with 50% PhiX.

The index tags were utilised to segregate sequencing reads for each sample. The total number of barcodes for each gene within the sample was counted using a Python script. Relative abundance was calculated for each mutant by dividing its barcode count by the total number of barcodes in the sample. The fold change in relative abundance between the mutant barcode in the mosquito midgut (output) and that in the mouse blood (input 2) was calculated and normalised with spike-in control barcodes for genes whose knockout did not exhibit a phenotype in the mosquito stages (KIN, PBLP, and PBANKA_0308200) to generate the oocyst conversion rate. The fold change in relative abundance between the mutant barcode in the

mouse blood (input 2) and that in the transfection cuvette (input 1) was used to calculate the off-target effect of gRNA for each gene (Fig. 3a). Normalised Barcode counts for each gene can be found in the source data file. Software and analysis details were deposited in GitHub (https://github.com/vpandey-om/HomingCas9) and Zenodo (https://zenodo.org/records/15073269).

### Generation of CRTL KO and tagged transgenic lines

The *Plasmo*GEM knockout and tagging vectors PbGEM-93381 and PbGEM-93389 were used to generate *crtl*⁻ and *crtl*-3xHA tagged parasites in Pb Bern mCherry and *P. berghei* ANKA cl15cy1 background, respectively. Briefly, *P. berghei* schizonts were electroporated with 3 mg of each plasmid using the FI115 programme on the Amaxa Nucleofector 4D (Lonza). Transfected parasites were promptly injected intravenously into BALB/c mice and were subjected to selection with 0.07 mg/mL of pyrimethamine in drinking water starting from day one post-infection. Clonal lines were derived by limited dilution cloning and were genotyped to verify modifications in the PBANKA_0916000 gene locus. Parasite lines and vectors produced by are available through the lead contact. Requests for *Plasmo*GEM vectors should be made through the resource (https://plasmogem.serve.scilifelab.se/pgem/).

### TEM

Mosquito midgut was fixed with 2.5% Glutaraldehyde (TAAB Laboratories, Aldermaston, England) in 0.1 M phosphate buffer. Samples were further post-fixed in 1% aqueous osmium tetroxide, dehydrated with ethanol and finally embedded in Spurr's resin (TAAB Laboratories, Aldermaston, England). All steps were performed using the Pelco Biowave pro+ (Ted Pella, Redding, CA). 70 nm ultrathin sections were picked up on copper grids and post-stained with 5% aqueous uranyl acetate and Reynolds lead citrate. Grids were examined with Talos L120C (FEI, Eindhoven, The Netherlands) operating at 120 kV. Micrographs were acquired with a Ceta 16 M CCD camera (FEI, Eindhoven, The Netherlands) using Velox ver 2.14.2.40.

### Reporting summary

Further information on research design is available in the Nature Portfolio Reporting Summary linked to this article.

## Data availability

Raw barcode sequence counts have been deposited in GitHub (https://github.com/vpandey-om/HomingCas9) and Zenodo (https://zenodo.org/records/15073269). Source data are provided with this paper.

## Code availability

Code for analysis has been deposited in GitHub (https://github.com/vpandey-om/HomingCas9) and Zenodo (https://zenodo.org/records/15073269).

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

## Acknowledgements

The authors are grateful to Sarah Lundgren, Divya Das, and Rashmi Mishra, who bred and prepared the mosquitoes for this work. SciLifeLab National Genomics Infrastructure in Stockholm is acknowledged for sequencing support. Kai Matuschewski and Manuel Rauch from Humboldt-Universität zu Berlin generously shared their PV1-mCherry line. The authors acknowledge Chiara Currà, Inga Siden-Kiamos (Foun-dation for Research and Technology—Hellas, Institute of Molecular Biology and Biotechnology), and Rita Tewari (University of Nottingham) for valuable input. We are grateful for technical assistance from the Umeå Core Facility for Electron Microscopy (UCEM) at the Chemical Biological Centre (KBC). Work at Umeå University received funding from the Knut and Alice Wallenberg Foundation and the European Research Council (Grant agreement No. 788516). MH was supported by SNF (P2SKP3_187635), HFSP (LT000131/2020-L), and a Marie Sklodowska-Curie Action fellowship (No. 895744). The National Microscopy Infra-structure NMI is supported by the Swedish Research Council (VR-RFI 2019-00217).

## Author contributions

Conceptualisation, O.B. and A.B.; Methodology, O.B., A.B. and M.H.; Software, V.P.; Formal Analysis, O.B., A.B., M.H., V.P., D.D.; Investigation, A.B., M.H., P.T., D.D.; Writing – Original Draft, A.B.; Writing – Review & Editing, O.B., M.H., D.D.; Visualisation, O.B., A.B., M.H., V.P., D.D.; Supervision, O.B., A.B.; Project Administration, A.B.

## Funding

## Competing interests

The authors declare no competing interests.
