## [Peer Review file · Nature Communications]

A CRISPR homing screen finds a chloroquine resistance transporter-like protein of the Plasmodium oocyst essential for mosquito transmission of malaria

Corresponding Author: Professor Oliver Billker

Version 0:

Reviewer comments:

Reviewer #1

(Remarks to the Author)

The author's lab previously established the barcoded PlasmogEM vector-based genetic screens and successfully identified critical genes in the haploid blood stage, gametocyte, and liver stage of P.b. However, a similar screen strategy could not be adapted to the parasite post-fertilization development and oocyst development because of the diploid or polyploidy statute. In this study, the author combined the Barcoded PlasmogEM KO vector with CRISPR/Cas9 and developed a homing screening strategy for detecting the genes critical in the oocyst of P.b. From a small-scale pilot screen, the authors revealed an essential function of previously described protein CRTL (chloroquine resistance transporter-like protein, PBANKA_0916000) in oocyst and sporozoites development.

Major concern:

>Throughout figures and supplementary figures, it is not reader-friendly. Some words are incompletely phrased. Many marks or labels in the figures are not explained or described in the figure legend. It is hard to follow for the reader with no detailed background. The authors should check the manuscript thoroughly and add all the information.

>What are the roles of gRNA1 and gRNA2 in both Figure 1 and Figure 2. The author did not describe it in the legend or the manuscript. It is hard to understand.

> Line 131-134 and Fig S1H. The cross between the fd1-cas9 clone and md4-cas9 clone produced much less oocyst (20 in average) compared to the wild type and other cross of parasites (200 in average). Since the cross between female-only md4-cas9 clone and male-only fd1-cas9 clone is the principal basis for the homing screen in this study, it brings the possibility (also a big concern) that the candidate genes critical in oocyst formation will be missed in the screen.

> The results in Figure 3b showed the homing screening results. Among these 21 genes tested, there are 7 genes with known or likely functions in the functionally diploid mosquito stages. Notably, 4 genes (DEH, 1019800, 1342500, and PBLP) were believed to be critical in oocyst or sporozoite development but did not show a difference between -homing or +homing group. Why did the homing screen system fail to detect them (4 out of 7)? This is a core question for the quality of the homing screen developed for this stage.

>Line 144-174, the cas9-based genome homing editing efficiency after fertilization was only evaluated using a reporter gene (color swapping). To quantitatively evaluate the genome editing efficiency, more direct evidence such as detecting the percentage of cells with target DNA editing is required. Gene homing test targeting the sequence of an endogenous gene is also necessary and preferred.

> Line 178-188, the reason that the author chooses the 21 genes in the pilot homing screen is not clear.

> In Fig 3B, how the oocyst conversion rates were calculated from barcode sequencing. There is no information provided. The authors should provide the formulas and codes used for data normalization and oocyst conversion rate calculation.

> Line 256-268, the author concluded the existence of hemozoin crystals in the oocyst, which is only evidenced by the results from the light microscopy observation in Figure 4. To confirm the ID of hemozoin crystals, other independent evidence is necessary.

> The author claimed the PBANKA_0916000 encodes a digestive vacuole protein. In Fig 4, a protein marker of the digestive vacuole should be used to confirm the digestive vacuole localization of CRTL protein by two-colored staining.

Minor concerns:

Line 172: Only two panels (A and B) in Fig 2. What is the Fig 2C?

Fig S1D: The ladder shown in this image of blot was trimmed and stitched.

Fig S1G: "Exflag<ellation" on Y-axis should be "Exflagellation".

Fig S2C: The molecular weight of DNA ladders massed, and the DNA ladders is missed in some images.

Fig S4D and S4E, wrong labeled with each other.

The authors mentioned that their raw array data and coding were deposited in github, but didn't provide any links or reference to their github repositories.

Reviewer #2

(Remarks to the Author)

This is an outstanding article by Balakrishnan, from the Billker lab in Swden, and colleagues. They have designed a very clever screen to generate homozygous knockouts in *P. berghei* in order to identify genes essential for oocyst formation and parasite development in *Anopheles* mosquito vectors. I have only a few comments and applaud the authors for very original and high-quality work.

Figure 1: For the homing vector column, when describing binding of the gRNA:Cas9 complex on the Zygote line, shouldn't the right side be gRNA2:Cas9 in green and not gRNA1:Cas9? I thought gRNA1 was the purple site to the left. Also, please specify in the text that the strategy should work in all cases of either gRNA1 alone, gRNA2 alone, or gRNA1+2, which each can lead to a double strand break and homology-directed recombination and repair.

Can the authors provide RNA data confirming lack of transcription of CRTL in *P. berghei* asexual blood stage parasites? Also preferably gametocytes? The IFA image in Figure S4d shows no signal in the CRTL-3xHA schizonts yet the positive signal in the ookinete is very small and overall the data do not appear as robust as one would wish to see. Additional IFA images would also be preferable, including asexual blood stage trophozoites and ideally male and female gametocytes.

When discussing the sequence and structural similarity between CRTL and CRT (Figures S7, S8), the authors should make it clear that the similarity with PfCRT is at the structural level, with the primary amino acid sequence similarity being fairly low. They should state % identity and similarity. They may also wish to add these values for *P. berghei* CRTL and *P. berghei* CRT.

Minor: the Figure 5 legend has a typo (should be "significant")

Reviewer #3

(Remarks to the Author)

Summary

This study introduces a novel genetic screening method for *Plasmodium* parasites using CRISPR-based homing vectors to reveal gene functions in diploid mosquito stages. The researchers developed single-sex parasite lines expressing Cas9 and demonstrated genome editing after fertilization. This innovation represents a modification of the highly successful PlasmoGEM approach and overcome one its important limitations, analyzing genes with essential functions in the multiploidy sexual stages of development in the mosquito. To demonstrate the potential scalability of this approach to systematically discover active malaria transmission genes after fertilization, a pilot screen of 21 genes were functionally characterized. Phenotypes of previously characterized genes were consistent with their previous characterizations as proof of concept. In addition, the screen identified PBANKA_0916000, encoding a chloroquine resistance transporter-like (CRTL) protein, as essential for oocyst development. Further characterization showed CRTL localizes to digestive vacuole-like structures in oocysts and is critical for their growth and sporulation. In silico structural analysis revealed PbCRTL and PfCRT have a high degree of structural similarity, including the DMT fold critical for its putative transport function. This finding confirmed that the oocyst DV is essential for completion of oocyst development and production of infectious sporozoites. Further studies are needed to understand the essential function of PbCRTL and the role of the DV in oocyst development. Overall, this study represents a significant advance in expanding the molecular toolbox for this important malaria model system.

Major Comments:

1. The report highlights some of the anomalies and limitations of this CRISPR/Cas9-based system. This includes the consistent reduction in oocyst numbers of the female-only md4-cas9 clone and loss of fitness for the G3PDH. These represent potential concerns on the validity of annotated phenotypes for at least some sexual stage genes given the

unknown basis of these variables. Sexual stage development is multigenic and mediated by a cascade of complex interactions that remain poorly characterized. Therefore, the basis of aberrant level of oocyst development of the female-only md4-cas9 clone is likely to alter phenotypes of some sexual stage genes. The lost functionality of G3PDH is presumed to be an artifact in cloning or in the selection process. Alternatively, CRISPR systems can have off-target effects that could confound results of genetic factors that have functional interactions with the variable factors. WGS data are needed to determine if there is identifiable genetic basis of these unexpected variations. Are these minor (e.g., SNPS) or major defects (e.g., large INDELS). Identifying the potential genetic basis will put these variables into context and support more accurate validation of observed phenotypes.

2. The study showed different homing efficiencies when gRNAs were expressed from male (89%) versus female (97%) gametes. Since this variation could impact the consistency of results across different experimental setups, some additional discussion on how this might impact analyzing different types of genes, such as targeting male vs female genes.

3. The exact timing of Cas9-mediated gene disruption after fertilization is not precisely known. This uncertainty could affect the interpretation of phenotypes, especially for genes with early post-fertilization functions. How do the authors plan to identify these genes? Some additional data would be helpful or at least provide a discussion on how timing might alter targeting of selected genes. This discussion should include how potential chromatin effects may alter target accessibility.

Minor Comments:

1. Provide more complete rationales for selection of all 21 genes of interest.

2. The study acknowledges that maternally inherited transcripts or proteins may mask phenotypes for some genes. This limitation could lead to false negatives in the screen, particularly for genes with products that persist long after fertilization. Are any steps taken during analysis to normalize the transcript numbers to account for this?

3. Cross-Species Validation? The study identifies a novel gene (CRTL) essential for oocyst development in *P. berghei* and infers the functional role of this gene based on sequence in other Plasmodium species and apparent structural similarity of other Plasmodium homologs, especially PfCRT. This comparative analysis is useful, but relatively superficial and not supported by additional experimental data. These should be included if additional data are available. If not, then more in-depth analysis of critical domains/residues should be included, especially for the DMT fold. Cross-species validation or conservation of the pilot screen genes would enhance the study and provide stronger support on the use of the rodent malaria model as a system to understanding human malaria parasite biology.

-related: move figure S8 to figure 6 of main text.

4. It would be beneficial to include expression of these 21 genes across various life cycle stages to enhance gene annotation efforts. Integrating these genetic screening hits with transcriptomics, proteomics, and metabolomics data could provide deeper insights.

5. How do the genes targeted influence vector compatibility?

6. In pooled screens, competition or interaction effects between different mutants might affect their relative abundances. These effects might not accurately reflect the phenotype of individual mutants in isolation. Have the authors addressed or observed such bias?

7. The authors repeatedly emphasized that this approach can be scaled, implying genome scale, even though 'only 21' genes were studied for this report. What are the practical limits on scalability?

8. If homing is not 100% efficient, some parasites might retain heterozygosity or mosaicism, complicating the interpretation of barcode abundances. Can the authors conclude that the differences in targeting efficiency are due to ploidy of the gametocytes?

9. For the CRTL knockout, genetic complementation experiments would provide stronger evidence that the observed phenotypes are specifically due to CRTL loss. Is that feasible?

10. How many mutants were created for each gene targeted?

11. While the study identifies CRTL as crucial for oocyst development, more mechanistic details on how it maintains vacuole acidification would strengthen the findings. It is possible that CRTL interacts with specific ion channels or transporters to regulate the proton gradient within the vacuole. Additionally, CRTL might be involved in the signaling pathways that control the activity of vacuolar ATPases, enzymes responsible for pumping protons into the vacuole to maintain its acidic environment. Can the authors discuss/speculate further on the functional physiology of the food vacuole?

12. The image sizes and resolution for Figs. 5c-f are inadequate. Fig 5a-b can be moved to the supplemental material if more space is needed to allow all the microscopic images to be enlarged. In addition, include hi-res images in the supplement, especially important for EM images.

13. The study does not explore CRTL's interactions with other proteins or its involvement in protein complexes, which may contribute to its role in oocyst development. Are there other transporter proteins in the oocyst that play crucial roles in development? How do these transporters interact with each other and with CRTL?

14. How valid are the alpha fold and other 3D models in Fig 6.

Reviewer #4

(Remarks to the Author)

Version 1:

Reviewer comments:

Reviewer #1

(Remarks to the Author)

In this revised version of the manuscript, the authors have addressed some of the comments.

I have three remaining points.

1> In Supplementary figure 3C, there are two main bands for CRTL-3HA protein in the immunoblot. One is the product with expected size, what is the other band? Unspecific protein or degraded product? It is better to describe them.

2> Throughout the figures, it is good practice to provide the experiment repeat information for each experiment in the legend. One, two, or three independent experiment performed with similar results?

3> Provide the raw image for all the immunoblot.

(Remarks on code availability)

Reviewer #2

(Remarks to the Author)

This is an excellent revision by Oliver Billker and coauthors. They provide a comprehensive rebuttal to the detailed reviews. My comments as Reviewer #2 were addressed well. I am fully satisfied with their revised manuscript and find this to be an excellent study for Nature Communications.

Minor:

Lines 205-7: To obtain accurate homing rates mosquitoes each mosquito included in the analysis... Can delete "mosquitoes" or state "in mosquitoes"

Also, can the authors specify whether AlphaFold predicts 10 transmembrane domains in CTRL. Are they predicted to be aligned mostly as antiparallel pairs as in PfCRT? It would seem so from Figure 6 but would be helpful to spell this out in the text.

(Remarks on code availability)

Reviewer #4

(Remarks to the Author)

The revised manuscript has adequately addressed the major and minor concerns raised in the initial review. The study presents a novel CRISPR homing screen to investigate Plasmodium oocyst-stage gene functions, an area of significant interest for malaria transmission research. The discovery of CRTL as a critical component of the oocyst digestive vacuole is a significant contribution.

The expanded discussion on gene selection rationale, homing efficiency differences between male and female gametes, and potential biases in pooled screens strengthens the study's impact. The integration of transcriptomic data and refined figures further enhance clarity.

(Remarks on code availability)

We thank the reviewers for their kind and constructive comments. We respond with new data, amended figures and substantial revisions in the text, which in the manuscript file is shown in blue to guide reviewers to relevant sections that where major edits were done. Minor edits (typos, references to figures) are not highlighted.

REVIEWER COMMENTS

Reviewer #1 (Remarks to the Author):

The author's lab previously established the barcoded PlasmoGEM vector-based genetic screens and successfully identified critical genes in the haploid blood stage, gametocyte, and liver stage of P.b. However, a similar screen strategy could not be adapted to the parasite post-fertilization development and oocyst development because of the diploid or polyploidy statute. In this study, the author combined the Barcoded PlasmoGEM KO vector with CRISPR/Cas9 and developed a homing screening strategy for detecting the genes critical in the oocyst of P.b. From a small-scale pilot screen, the authors revealed an essential function of previously described protein CRTL (chloroquine resistance transporter-like protein, PBANKA_0916000) in oocyst and sporozoites development.

Major concern:

>Throughout figures and supplementary figures, it is not reader-friendly. Some words are incompletely phrased. Many marks or labels in the figures are not explained or described in the figure legend. It is hard to follow for the reader with no detailed background. The authors should check the manuscript thoroughly and add all the information.

Thank you for the feedback. We have now improved the manuscript by adding details and explanations to the figure legend to explain our findings better.

What are the roles of gRNA1 and gRNA2 in both Figure 1 and Figure 2. The author did not describe it in the legend or the manuscript. It is hard to understand.

Each homing vector was designed to express two gRNAs simultaneously to reduce the risk of false negatives if one gRNA works with low efficiency. We now explain this better in the legend of Fig. 1.

> Line 131-134 and Fig S1H. The cross between the fd1-cas9 clone and md4-cas9 clone produced much less oocyst (20 in average) compared to the wild type and other cross of parasites (200 in average). Since the cross between female-only md4-cas9 clone and male-only fd1-cas9 clone is the principal basis for the homing screen in this study, it brings the possibility (also a big concern) that the candidate genes critical in oocyst formation will be missed in the screen.

We shared the reviewers concerns. Since we wanted to mutagenised the male parent for the screen in order not to lose female fertility genes from the screen, we chose to cross the mutagenised *fd1-cas9* line with a fully fertile Cas9 expressing line. We now see that we did not explain this well. This decision allowed us to avoid the low fertility of the *md4-cas9* clone. We obtained 400 oocysts per midgut on average. Around half of these will have resulted from selfing of the *cas9* line, but since cysts from selfing have no barcodes, they are not visible in the screen. We have now included additional text in the manuscript and figure legend to explain this better.

The results in Figure 3b showed the homing screening results. Among these 21 genes tested, there are 7 genes with known or likely functions in the functionally diploid mosquito stages. Notably, 4 genes (DEH, 1019800, 1342500, and PBLP) were believed to be critical in oocyst or sporozoite development but did not show a difference between -homing or +homing group.

The reviewer is right to expect mutants with known oocyst phenotypes to be informative for the reliability of our approach to uncover oocyst biology. We should have given more room to these mutants to explain why the screen actually performed very well. We have now expanded the results text to explain this better. We also added information in Table S1 (column L), where we evaluate for each gene whether the screen result is consistent with prior knowledge (referenced in column N). Importantly, we find that only one screen result, that for *pank2*, is inconsistent with published functional data on a gene.

Three of the known oocyst genes produced hits in the screen. The transcription factor (*ap2-o4*), a glutathione reductase (*gr*) and the secreted ookinete adhesive protein (*soap*). For pantothenate kinase 2 (*pank2*), a complete loss of oocysts was expected (Srivastava et al., 2016), but the homing effect we observe is smaller than for other known oocyst genes (1.7-fold), and does not reach statistical significance. This could mean that homing was not efficient enough in this mutant, or that sufficient PANK2 protein or its enzymatic product was inherited from female gametocytes to overcome the essential oocyst function of this enzyme. The fifth oocyst gene is a 3-hydroxy acyl-CoA dehydratase (*deh*), the deletion of which was previously shown not to reduce oocyst numbers but to cause their gradual degeneration from day 7 (Guttery et al., 2020, 2014). The lack of a homing effect for this gene is consistent with the fact that the degenerating oocysts would still contain barcodes.

Importantly, published phenotypes of mutants in PBANKA_1019800 (*cdlk*), PBANKA_1342500 KO (*smp3*) and PBANKA_0712200 KO (*pblp*) have similar numbers of replicating oocyst as wild type (see references now provided in Table. S1) and would not therefore show a loss of oocyst barcodes. Had we chosen salivary gland sporozoites as the endpoint for barcode counting, these mutants and *deh* may well have revealed a phenotype. In summary, it appears that the screen recapitulates most, but not all published phenotypes at the oocyst stage, and that in the case of *deh*, we probably see no effect because of the late function of this gene.

We have now expanded the text to explain this better and refer to the Supplementary Table 1.

We find the homing method robust enough to be scaled up to define gene functions in oocysts more systematically than has been possible so far. Although such a screen will probably miss some relevant genes, it can be said with confidence that it will detect many new gene functions. To reveal late oocyst gene functions that the current design is unable to detect, a future screen should include a salivary gland endpoint.

Line 144-174, the cas9-based genome homing editing efficiency after fertilization was only evaluated using a reporter gene (color swapping). To quantitatively evaluate the genome editing efficiency, more direct evidence such as detecting the percentage of cells with target DNA editing is required. Gene homing test targeting the sequence of an endogenous gene is also necessary and preferred.

We agree that homing needs to be tested on an endogenous gene in its normal chromosomal context, which provides the homology for repair. The colour swapping experiments were in fact not done with a transgene, but we inserted the green and red fluorescent C-terminal tags into the endogenous *myoA* locus. We also targeted the gRNAs to endogenous *myoA* sequences. The only artificial part is that the endogenous gene was tagged to provide a microscopically quantifiable readout at the level of live single oocysts. We now explain this better in the manuscript.

We further agree that we have not shown at the molecular level how the second copy of a target gene is disrupted, but we show that colour swapping relies on both Cas9 and the guide RNA for the target locus. We know the target locus must not only get cleaved but also repaired, because an unrepaired double strand break would impart a severe fitness cost irrespective of the function of the target gene. Perhaps one could argue that repair in our system may not be homology directed, but this is highly unlikely since the loss of the non-homologous end joining machinery from *Plasmodium* genomes is known to have led to a reliance on homology-directed repair. We acknowledge, however, that our interpretation infers some knowledge from more tractable systems and have added a paragraph in the discussion defining what we do and do not know about the system.

We thought hard about ways to investigate at the level of the DNA and for many endogenous genes simultaneously how efficiently homing happens with different targets. Importantly, the genomes of sporozoites and subsequent stages would bear no trace of the homing. It should be possible to monitor homing efficiency by qPCR or Southern hybridisation in populations after mosquito transmission, but this would only work for one mutant at a time and with genes where there is no selection for or against the disrupted locus, i.e. with genes that have no function between fertilisation and whenever one chooses to quantify alleles. If that time point is the mature oocyst, qPCR is the only option and may be difficult due to the small number of parasites. Subsequent blood stages would be easier to analyse, but harder to avoid selection effects. Either way, there are few genes we can be sure have no function except perhaps members of multigene families whose subtelomeric location and epigenetic

regulation may bias targeting efficiency. Either way, these were not the genes we wanted to study. We therefore opted instead to measure phenotype as a proxy for efficiency across a panel of genes, having first shown good efficiency for one tagged endogenous locus.

Line 178-188, the reason that the author chooses the 21 genes in the pilot homing screen is not clear.

Most genes were selected to ask if the screen design produced an outcome that was consistent with the known knock-out phenotype. At least three target genes with known functions for each life cycle stage were included, as now explained better in the manuscript. The reasoning for including each gene is given in Supplementary Table 1. To validate the screen with genes of unknown function we also included a few genes expressed highly in the oocyst according to the Malaria Cell Atlas, which resulted in the discovery of *ctrl*. In response to reviewer 2, we have now included expression data of all selected genes as Supplementary Fig. 4.

In Fig 3B, how the oocyst conversion rates were calculated from barcode sequencing. There is no information provided. The authors should provide the formulas and codes used for data normalization and oocyst conversion rate calculation.

This was described in general terms in the methods. We now also provide a formula. We also point to a Python script on GitHub which we created for barcode counting and to perform calculations (<https://github.com/vpandey-om/HomingCas9>).

> Line 256-268, the author concluded the existence of hemozoin crystals in the oocyst, which is only evidenced by the results from the light microscopy observation in Figure 4. To confirm the ID of hemozoin crystals, other independent evidence is necessary.

We did not mean to imply that the presence of malaria pigment in oocysts is a new discovery. These granules are, in fact, a well-known and defining feature of the oocyst. Relevant literature is now cited. It is clear from biochemical characterisation that in blood stages that pigment granules are made of hemozoin. The pigment granules of all parasite stages, including ookinetes and oocyst are similar in number and have the same refractile properties when viewed with polarised or phase contrast illumination, and that these arise from the crystalline form of the hemozoin pigment. It is therefore likely that malaria pigment in oocysts still has the same crystal form as in gametocytes and ookinetes for which there is much EM evidence. The reviewer is correct to point out that we cannot be sure the hemozoin pigment of the late oocyst has retained its crystal shape from earlier stages, unless we can image the crystals. We now no longer equate pigment granules with hemozoin in the results section since we cannot think of a good way of verifying the biochemical nature of the granules from the oocyst stage, which cannot be purified in sufficient quantities for biochemical analysis.

Microscopically, oocyst pigment has to our knowledge only been visualised in pre-sporogonic oocysts. Canning and Sinden (1973) write of *P. berghei* 2-4 day cysts that "...Pigment granules were frequently seen ... as electron-dense granules of

various shapes enclosed by a single membrane(Canning & Sinden, 1973).” Pastrana-Mena et al. (2010) serendipitously sectioned a pigment-containing vacuole in 6-day-old oocysts of a glutathione reductase mutant showing crystals(Pastrana-Mena et al., 2010). In sporulating cysts, we have not found hemozoin crystals and we cannot see them in published transmission EM images from these late stages, although the granules clearly persist by light microscopy of whole cysts. We find it most likely that the crystals are lost from the sections during processing for transmission EM. Canning and Sinden (1973) have reported this for their ookinete preparations(Canning & Sinden, 1973).

Importantly, the precise form that malaria pigment assumes in the sporulating oocyst is not central to our conclusions. Acknowledging the uncertainty, we now refer to pigment granules, rather than hemozoin crystals. We have also added more text and references to provide the reader with additional background.

>The author claimed the PBANKA_0916000 encodes a digestive vacuole protein. In Fig 4, a protein marker of the digestive vacuole should be used to confirm the digestive vacuole localization of CRTL protein by two-colored staining.

The DV of oocysts has not previously received much attention and no canonical markers are available. We added new data to Fig. S5, illustrating that the oocyst localisation of CRTL is the same as that of the PV1 protein. Named originally for its location in the parasitophorous vacuole, PV1 is also present in the pigment-containing food vacuole of schizonts(Matz & Matuschewski, 2018), providing a connection to the corresponding location in oocysts. We now confirm the observation by Matz and Matuschewski (2018) that also in our hands, PV1 in mature oocysts colocalises with pigment granules. We tried to visualise PV1 and CRTL in the same cell after crossing the tagged lines, but unfortunately we did not manage to find conditions for immunohistochemistry that labelled both proteins simultaneously. We therefore show labelling in different cysts.

Minor concerns:

Line 172: Only two panels (A and B) in Fig 2. What is Figure 2C?

We have corrected the error.

Fig S1D: The ladder shown in this image of blot was trimmed and stitched.

The ladder was from the same blot that also contained irrelevant lanes. We now show the entire blots, including the irrelevant lanes.

Fig S1G: “Exflag<ellation” on Y-axis should be “Exflagellation”.

We have fixed the mistake.

Fig S2C: The molecular weight of DNA ladders massed, and the DNA ladders is missed in some images.

We have now included images with molecular weight ladders.

Fig S4D and S4E, wrong labeled with each other.

We have edited the mistake in the manuscript.

The authors mentioned that their raw array data and coding were deposited in github, but didn't provide any links or reference to their github repositories.

The link for github is now included in the manuscript (<https://github.com/vpandey-om/HomingCas9>).

Reviewer #2 (Remarks to the Author):

This is an outstanding article by Balakrishnan, from the Billker lab in Sweden, and colleagues. They have designed a very clever screen to generate homozygous knockouts in *P. berghei* in order to identify genes essential for oocyst formation and parasite development in *Anopheles* mosquito vectors. I have only a few comments and applaud the authors for very original and high-quality work.

Figure 1: For the homing vector column, when describing binding of the gRNA:Cas9 complex on the Zygote line, shouldn't the right side be gRNA2:Cas9 in green and not gRNA1:Cas9? I thought gRNA1 was the purple site to the left. Also, please specify in the text that the strategy should work in all cases of either gRNA1 alone, gRNA2 alone, or gRNA1+2, which each can lead to a double strand break and homology-directed recombination and repair.

Thank you for pointing it out. We have now changed the figure and specified the strategy in the figure legend.

Can the authors provide RNA data confirming lack of transcription of CRTL in *P. berghei* asexual blood stage parasites? Also preferably gametocytes? The IFA image in Figure S4d shows no signal in the CRTL-3xHA schizonts yet the positive signal in the ookinete is very small and overall the data do not appear as robust as one would wish to see. Additional IFA images would also be preferable, including asexual blood stage trophozoites and ideally male and female gametocytes.

We now show single-cell RNA seq data from the Malaria Cell Atlas, (Supplementary Fig. 5d) demonstrating that transcript abundance for CRTL peaks in the ookinete and persists in the oocyst. We have also expanded the IFA panel (Supplementary Fig. 5e) with new data shown absence of protein labelling from trophozoites, and male and female gametocytes.

When discussing the sequence and structural similarity between CRTL and CRT (Figures S7, S8), the authors should make it clear that the similarity with

PfCRT is at the structural level, with the primary amino acid sequence similarity being fairly low. They should state % identity and similarity. They may also wish to add these values for *P. berghei* CRTL and *P. berghei* CRT.

We have now included this information as follows:

“The similarity between *Pb*CRTL and *Pf*CRT is mainly structural. *Pf*CRT (424 amino acids) aligns with the C-terminus of *Pb*CRTL (1113 amino acids), showing 18.1% identity and 32.7% similarity in their amino acid sequences. The N-terminus of *Pb*CRTL, which is rich in asparagine and lysine, does not form part of the predicted transporter domain. In contrast, the syntenic orthologs between *P. berghei* and *P. falciparum* are much more alike, with 65.2% identity and 83.6% similarity in the case of CRT and 34.7% identity and 47.1% similarity for CRTL.”

Minor: the Figure 5 legend has a typo (should be “significant”)

We have changed it. Thank you.

Reviewer #3 (Remarks to the Author):

Summary

This study introduces a novel genetic screening method for Plasmodium parasites using CRISPR-based homing vectors to reveal gene functions in diploid mosquito stages. The researchers developed single-sex parasite lines expressing Cas9 and demonstrated genome editing after fertilization. This innovation represents a modification of the highly successful PlasmoGEM approach and overcome one its important limitations, analyzing genes with essential functions in the multiploidy sexual stages of development in the mosquito. To demonstrate the potential scalability of this approach to systematically discover active malaria transmission genes after fertilization, a pilot screen of 21 genes were functionally characterized. Phenotypes of previously characterized genes were consistent with their previous characterizations as proof of concept. In addition, the screen identified PBANKA_0916000, encoding a chloroquine resistance transporter-like (CRTL) protein, as essential for oocyst development. Further characterization showed CRTL localizes to digestive vacuole-like structures in oocysts and is critical for their growth and sporulation. In silico structural analysis revealed *Pb*CRTL and *Pf*CRT have a high degree of structural similarity, including the DMT fold critical for its putative transport function. This finding confirmed that the oocyst DV is essential for completion of oocyst development and production of infectious sporozoites. Further studies are needed to understand the essential function of *Pb*CRTL and the role of the DV in oocyst development.

Overall, this study represents a significant advance in expanding the molecular toolbox for this important malaria model system.

Major Comments:

1. The report highlights some of the anomalies and limitations of this

CRISPR/Cas9-based system. This includes the consistent reduction in oocyst numbers of the female-only *md4-cas9* clone and loss of fitness for the G3PDH. These represent potential concerns on the validity of annotated phenotypes for at least some sexual stage genes given the unknown basis of these variables. Sexual stage development is multigenic and mediated by a cascade of complex interactions that remain poorly characterized. Therefore, the basis of aberrant level of oocyst development of the female-only *md4-cas9* clone is likely to alter phenotypes of some sexual stage genes. The lost functionality of G3PDH is presumed to be an artifact in cloning or in the selection process. Alternatively, CRISPR systems can have off-target effects that could confound results of genetic factors that have functional interactions with the variable factors. WGS data are needed to determine if there is identifiable genetic basis of these unexpected variations. Are these minor (e.g., SNPS) or major defects (e.g., large INDELS). Identifying the potential genetic basis will put these variables into context and support more accurate validation of observed phenotypes.

The reduced transmissibility of the *md4-cas9* line and the failure to recover a G3PDH mutant are unrelated. Recognising the reduced fertility of the *md4-cas9* line, we chose not to use this line in the screen. As explained in response to reviewer 1, we crossed the *fd1-cas9* line with a fully fertile line constitutively expressing Cas9, which gave us 400 oocysts per midgut on average. Around half of these oocysts will have arisen from selfing of the *cas9* line, but since cysts from selfing have no barcodes, they are not visible in the screen. We apologise for not explaining this well and have now included additional text in the manuscript and figure legend.

Plasmodium parasites have lost the canonical non-homologous end joining (NHEJ) pathway to repair double strand breaks. Although *P. falciparum* possesses a non-canonical end joining mechanism, this is much less efficient than homology-directed repair (Kirkman et al., 2014). In ring stages, which have only one copy of the genome, a gRNA with off-target effects would therefore result in the death of the parasite, because no template for homology-directed repair is available. We believe this is what happens with the G3PDH mutant when the vector contains gRNAs. We see parasite death whenever we transfect an on-target gRNA without a repair template. Unlike in organisms with NHEJ, we would not expect to find SNPs or INDELS in a CRISPR screen in *Plasmodium*. This also means that if a mutant can be created easily at the blood stage, as evidenced by its barcode being abundant in the input pool for mosquito transmission, it is difficult to imagine how its gRNAs could have significant off-target effects in oocysts. We interpret the fact that the presence of gRNAs only affected the recovery of one of the mutants, G3PDH, as indication that off-target effects are not going to get in the way of scaling up the homing screen. We will redesign the G3PDH gRNAs individually in future to see if the effect can be linked to one of them.

2. The study showed different homing efficiencies when gRNAs were expressed from male (89%) versus female (97%) gametes. Since this variation could impact the consistency of results across different experimental setups, some additional discussion on how this might impact analyzing different types of genes, such as targeting male vs female genes.

We now discuss the opportunities and limitations of the system further in the 2nd and 3rd paragraph of the discussion. Importantly, even the apparently lower homing efficiency of delivering gRNAs through the male gamete allowed us to identify new gene functions. A main difference between mutagenising the male or the female parent is in the genes that can be studied. To use homing to look beyond a gene's first function in sex-specific fertility, a gene needs to be disrupted in the parent where the gene loss will not cause infertility. To look at mitochondrial functions, this requires mutagenising the male. If one wanted to look for later functions of axonemal dyneins that are essential for male fertility, one would need to mutagenise the parent of the female gamete.

Another complication we now discuss more deeply is related to this reviewer's next comment, namely how the timing of gene disruption affects phenotype. This is complicated by different layers of post-transcriptional gene regulation in the female gamete and oocyte. Female gametocytes transcribe and store mRNAs necessary for the development of zygotes and oocytes in a silenced form (Mair et al., 2006). At a later developmental stage both male and female genes may transcribe the same gene. Which sex is disrupted first will have different consequences.

3. The exact timing of Cas9-mediated gene disruption after fertilization is not precisely known. This uncertainty could affect the interpretation of phenotypes, especially for genes with early post-fertilization functions. How do the authors plan to identify these genes? Some additional data would be helpful or at least provide a discussion on how timing might alter targeting of selected genes. This discussion should include how potential chromatin effects may alter target accessibility.

We have now explained this better in the discussion. Working out when precisely a gene gets disrupted after fertilisation will be technically challenging. Either way, as with any stage-specific gene knockout, inherited protein and mRNA will affect when the phenotype can be seen. Genes that function early after fertilisation often introduce their products into the zygote with the female gamete. Such genes were identified in our recent fertility screen (Sayers et al., 2024). For instance, genes involved in meiotic recombination are important for female fertility, not for male fertility, either because the proteins come from the female gamete or only the female genome is expressed immediately after fertilisation, possibly because the highly compressed genome of the male gamete remains epigenetically silenced for longer (Bushell et al., 2009).

We do not know whether homing efficiency varies between genes but factors such as chromatin state of the target locus may play a role. These limitations have to be considered when interpreting the homing screen result. For this study, we have not verified the precise timing of homing and we have been unable to think of a good way of doing this. We suspect homing happens during meiosis, but if it occurs much later, this would interfere with revealing the function of genes essential for the development of the zygote.

Minor Comments:

1. Provide more complete rationales for selection of all 21 genes of interest.

Most genes in the screen were selected to have known first essential functions at different life cycle stages. The rationale was to reveal any major problems of the protocol with false positive and false negative hits, although we recognise larger numbers of known genes will need to be studied to calculate these rates accurately. The pragmatic approach was to select at least three target genes for each life cycle stage. We now explain this better in the manuscript. The reasoning behind gene selection is included in Table S1. To see if the screen could lead to new discoveries, we also included genes with unknown function whose mRNA is particularly abundant in the oocyst according to the Malaria Cell Atlas. We have also now included expression data of all selected genes as Supplementary Fig. 4.

2. The study acknowledges that maternally inherited transcripts or proteins may mask phenotypes for some genes. This limitation could lead to false negatives in the screen, particularly for genes with products that persist long after fertilization. Are any steps taken during analysis to normalize the transcript numbers to account for this?

If the absence of a phenotype in the screen reflects the complicated biology of the parasite, we would consider this a true negative. False negatives would be due to the absence of homing, for instance, i.e. a technical failure of the screen, or a poor signal-to-noise ratio when too few mosquitoes are dissected or barcodes are not sampled in a representative manner,

If a gene has an earlier than expected first essential function because a critical amount mRNA is stored already in the female gametocyte, then this can be detected in a different type of screen. We have just published a fertility screen that shows such earlier gene functions (Sayers et al., 2024). The homing design we report in the current manuscript, where the female parental gene copy remains intact until after fertilisation, is complementary because it allows us to look at the importance of *de novo* gene expression after fertilisation. We agree that integrating data from different screens with gene and protein expression data will ultimately provide a comprehensive view of how gene function and regulation are connected. The current dataset is too small to do this, but when scaling up screening in the mosquito stages we will keep this in mind.

3. Cross-Species Validation? The study identifies a novel gene (CRTL) essential for oocyst development in *P. berghei* and infers the functional role of this gene based on sequence in other Plasmodium species and apparent structural similarity of other Plasmodium homologs, especially PfCRT. This comparative analysis is useful, but relatively superficial and not supported by additional experimental data. These should be included if additional data are available. If not, then more in-depth analysis of critical domains/residues should be included, especially for the DMT fold. Cross-species validation or conservation of the pilot screen genes would enhance the study and provide stronger support on the use of the rodent malaria model as a system to

understanding human malaria parasite biology.
-related: move figure S8 to figure 6 of main text.

Allelic exchange experiments that express *P. falciparum* genes in *P. berghei* have been done by us and others in a few dozen cases with remarkable success, but almost always with a specific translational aim that required the *P. falciparum* ortholog to function in *P. berghei*, so that vaccine candidates could be validated, or drug and vaccine efficiency modelled *in vivo* (Bauza et al., 2013; Kolli et al., 2021; Longley et al., 2015; Manzoni et al., 2017).

We have not done cross-species complementation experiments for CRTL because the validity of our conclusions do not rest on PfCRTL being functional in *Plasmodium berghei*. Failure to complement would not rule out an identical function, because the inability of PfCRTL to function in *P. berghei* could be due, for instance, to very subtle structural differences affecting interactions with other proteins.

Cross species complementation is also not necessary to validate *P. berghei* as a model for *P. falciparum* because there many orthologs are known to have similar functions in both species, and the power of *P. berghei* to study genes in the mosquito stages is not in doubt. All genes we included in the pilot screen have syntenic orthologs in *P. falciparum*, none of which have been studied in the oocyst. Once genetic screens in *P. berghei* have created a more complete picture of oocyst biology, we will be in a better position to decide which aspects to validate in *P. falciparum* and which drug or vaccine targets to model by cross-species complementation.

We have now expanded Supplementary Table 1 to include the gene IDs of *P. falciparum* orthologs and what we know about their functions. We have also now moved Figure S8 to Figure 6.

4. It would be beneficial to include expression of these 21 genes across various life cycle stages to enhance gene annotation efforts. Integrating these genetic screening hits with transcriptomics, proteomics, and metabolomics data could provide deeper insights.

We agree. Integrating phenotype and expression data in this way will indeed be very powerful. We have just released a database in which we integrate data from our previous screens in this way and also enable visualisation of expression data. <https://plasmogem.serve.scilifelab.se/pgem/phenotype>

With the results from many screens it is now possible to cluster genes by function and generate new hypotheses. For quick reference, we have now included published single-cell transcriptomics data for all the genes on the screen as Supplementary Fig.4. Before we include data from a homing screen in our database and PlasmoDB, we want to scale up the approach.

5. How do the genes targeted influence vector compatibility?

This is a very interesting question to explore. We find it likely that future comparative homing screens, for instance in *A. gambiae* vs. *A. stephensi* or in immune vs. non-immune mosquitoes, will reveal vector-parasite genetic interactions that determine vectorial competence. However, in the absence of a good hypothesis why CRTL might be important for vector interactions, our emphasis will be on scaling up the screening approach to explore this question systematically.

6. In pooled screens, competition or interaction effects between different mutants might affect their relative abundances. These effects might not accurately reflect the phenotype of individual mutants in isolation. Have the authors addressed or observed such bias?

From our published screens we know that there are differences between studying mutants in pools vs. individually. Most are related to our ability to measure biological parameters. Detecting small differences in growth or stage conversion rates between mutants is much easier when mutants can be compared directly within the same host. On the other hand, some mutants in pools are rare, which increases variance between replicates and thereby reduces our power to detect phenotypes. We consider these factors in our experimental design and statistical analysis.

Beyond these statistical factors, there are bound to be some profound biological interactions between mutants which would be exciting to detect. One can for instance imagine a scenario where oocysts manipulates the mosquito environment in a manner that is not cell-autonomous, for instance through a secrete a factor. A pooled screen may miss such a factor. For the alternative scenario, where a single mutant would have a dominant effect on the rest of the pool, we find it harder to think of a mechanism. When we scale up the homing screen, we will still need to screen in pools for statistical reasons. We will include a small number of control mutants into each pool to allow between-pool comparison, which may detect any dominant effects. In the current screen, we have not verified this since the pool was quite small and dominant effects are probably rare.

7. The authors repeatedly emphasized that this approach can be scaled, implying genome scale, even though 'only 21' genes were studied for this report. What are the practical limits on scalability?

Scaling up would first require the insertion of gRNAs into each of around 1300 *PlasmoGEM* vectors that target genes known not to be essential in the blood stages. Given the overall coverage of the *PlasmoGEM* resource, this should amount to around 65% of all targetable genes of the core genome. Doing this at scale in 96-well plates, we would have to accept failing with some of the vectors due to diminishing returns. How many genes can be studied in one pool is limited by transfection efficiency and by how many oocysts can be produced by mosquitoes feeding on the same mouse. In our experience it would be necessary to screen 50-100 mutants per pool in triplicate at oocyst and salivary gland sporozoite stage. The screen would require dissecting 4500 midguts and the same number of pairs of glands. The screen itself would be the work of a postdoc and a research assistant for two years. We are currently raising funding for this work.

8. If homing is not 100% efficient, some parasites might retain heterozygosity or mosaicism, complicating the interpretation of barcode abundances. Can the authors conclude that the differences in targeting efficiency are due to ploidy of the gametocytes?

The reviewer is right to suspect that homing will probably not be efficient for every gene, which means we miss phenotypes. A screen would need to be infinitely robust to reveal every small genetic effect on phenotype. We therefore do not ask whether a screen would detect everything but whether it will find enough to justify the effort and move the field forward. We know that caution needs to be exercised when interpreting negatives, especially with a technically challenging screen like this one, but above all we need to be confident that hits are robust enough to justify testing the hypotheses that they generate. Inefficient homing will mean less power to detect small effect sizes. The only way to find out if we have a viable protocol is to run the type of pilot experiment this manuscript is about. We suspect some of the smaller effect sizes may be down to imperfect homing, but we also conclude the overall picture is very promising, and CRTL was discovered and validated robustly from among a small number of genes of unknown function.

9. For the CRTL knockout, genetic complementation experiments would provide stronger evidence that the observed phenotypes are specifically due to CRTL loss. Is that feasible?

Complementation or analysing two independently produced clones are considered standard in *P. berghei* to avoid cloning artefacts in linking a genetic modification to a phenotype. In the case of CRTL, the screen phenotype is not from a clone but many independent genetic modification events. It is corroborated by the cloned knock out. We are therefore confident the phenotype is linked to the genetic modification. The phenotype is further consistent with the stage specific expression of the protein. We also find a good correlation between protein localisation (DV) and aspects of the phenotype (DV swelling).

One could consider ruling out that the CRTL KO phenotype originates from disturbing neighbouring genes by complementation in trans, i.e. inserting the CRTL gene into another locus under its own control elements. Although we have not previously encountered neighbouring gene effects, we looked at the vector design with this in mind and also considered that the tagging vector did not produce a phenotype. Complementation in trans is not generally considered a necessary control in the field, and we find nothing that would suggest the CRTL mutant needs to be complemented in this particular way.

10. How many mutants were created for each gene targeted?

The pool for the screen was only created once, but each type of mutant within the pool would have arisen many times from dozens or hundreds of independent events and was controlled against the remaining mutants in the same pool, ruling out cloning artefacts. The homing screen was replicated and homing happened independently in each zygote. The CRTL phenotype was validated with a single

clone, but data from pool and clone are entirely consistent (see also previous response).

11. While the study identifies CRTL as crucial for oocyst development, more mechanistic details on how it maintains vacuole acidification would strengthen the findings. It is possible that CRTL interacts with specific ion channels or transporters to regulate the proton gradient within the vacuole. Additionally, CRTL might be involved in the signaling pathways that control the activity of vacuolar ATPases, enzymes responsible for pumping protons into the vacuole to maintain its acidic environment. Can the authors discuss/speculate further on the functional physiology of the food vacuole?

While we here report for the first time, to our knowledge, that the oocyst DV is acidified, we do not claim that CRTL is directly responsible. We have expanded the discussion by pointing to the known mechanism of DV acidification in *P. falciparum* blood stages. The identification of CRTL now provides an opportunity to explore the role of this acidic compartment of the oocyst further.

12. The image sizes and resolution for Figs. 5c-f are inadequate. Fig 5a-b can be moved to the supplemental material if more space is needed to allow all the microscopic images to be enlarged. In addition, include hi-res images in the supplement, especially important for EM images.

We have increased figure sizes for these in the main figure. Additionally, we have included enlarged EM images in the Source Data file.

13. The study does not explore CRTL's interactions with other proteins or its involvement in protein complexes, which may contribute to its role in oocyst development. Are there other transporter proteins in the oocyst that play crucial roles in development? How do these transporters interact with each other and with CRTL?

Protein interactions of CRTL would indeed be nice to know. Two other proteins of the oocyst DV have been identified, PV1 (Matz & Matuschewski, 2018) and MDR2 (Rijpma et al., 2016), but their molecular functions are also unknown. We now mention these in the text. Unfortunately, we do not have a method to do protein pulldowns from infected midguts. The small amount of biological material that can be produced has so far prevented biochemical studies on oocyst.

14. How valid are the alpha fold and other 3D models in Fig 6.

We have now included a panel in Figure 6 to show the quality scores of the predicted CRTL structure. The alphaFold model of CRTL is based on an experimentally determined structure of CRT. The core DMT fold of CRTL, which aligns well with the structure of PfCRT, has a predicted local distance difference test (pLDDT) score of greater than 90 in the alphaFold model showing a very high confidence in the transporter region. Interestingly, CRTL has an unstructured N-terminal extension and many predicted disorder regions with low pLDDT scores of 30-50 suggesting

that these regions could have additional functions or might be relevant to recruit other proteins to CRTL.

Reviewer #4 (Remarks to the Author):

References

- Bauza, K., Malinauskas, T., Pfander, C., Anar, B., Jones, E. Y., Billker, O., Hill, A. V. S., & Reyes-Sandoval, A. (2013). Efficacy of a *Plasmodium vivax* malaria vaccine using ChAd63 and modified vaccinia Ankara expressing thrombospondin-related anonymous protein as assessed with transgenic *Plasmodium berghei* parasites. *Infection and Immunity*, *82*(3), 1277–1286.
- Bushell, E. S. C., Ecker, A., Schlegelmilch, T., Goulding, D., Dougan, G., Sinden, R. E., Christophides, G. K., Kafatos, F. C., & Vlachou, D. (2009). Paternal effect of the nuclear formin-like protein MISFIT on *Plasmodium* development in the mosquito vector. *PLoS Pathogens*, *5*(8), e1000539.
- Canning, E. U., & Sinden, R. E. (1973). The organization of the ookinete and observations on nuclear division in oocysts of *Plasmodium berghei*. *Parasitology*, *67*(1), 29–40.
- Guttery, D. S., Pandey, R., Ferguson, D. J., Wall, R. J., Brady, D., Gupta, D., Holder, A. A., & Tewari, R. (2020). Plasmodium DEH is ER-localized and crucial for oocyst mitotic division during malaria transmission. *Life Science Alliance*, *3*(12), e202000879.
- Guttery, D. S., Poulin, B., Ramaprasad, A., Wall, R. J., Ferguson, D. J. P., Brady, D., Patzewitz, E.-M., Whipple, S., Straschil, U., Wright, M. H., Mohamed, A. M. A. H., Radhakrishnan, A., Arold, S. T., Tate, E. W., Holder, A. A., Wickstead, B., Pain, A., & Tewari, R. (2014). Genome-wide functional analysis of *Plasmodium* protein phosphatases reveals key regulators of parasite development and differentiation. *Cell Host & Microbe*, *16*(1), 128–140.
- Kirkman, L. A., Lawrence, E. A., & Deitsch, K. W. (2014). Malaria parasites utilize both homologous recombination and alternative end joining pathways to maintain genome integrity. *Nucleic Acids Research*, *42*(1), 370–379.
- Kolli, S. K., Salman, A. M., Ramesar, J., Chevalley-Maurel, S., Kroeze, H., Geurten, F. G. A., Miyazaki, S., Mukhopadhyay, E., Marin-Mogollon, C., Franke-Fayard, B., Hill, A. V. S., & Janse, C. J. (2021). Screening of viral-vectored *P. falciparum* pre-erythrocytic candidate vaccine antigens using chimeric rodent parasites. *PLoS One*, *16*(7), e0254498.
- Longley, R. J., Salman, A. M., Cottingham, M. G., Ewer, K., Janse, C. J., Khan, S. M., Spencer, A. J., & Hill, A. V. S. (2015). Comparative assessment of vaccine vectors encoding ten malaria antigens identifies two protective liver-stage candidates. *Scientific Reports*, *5*, 11820.
- Mair, G. R., Braks, J. A. M., Garver, L. S., Wiegant, J. C. A. G., Hall, N., Dirks, R. W., Khan, S. M., Dimopoulos, G., Janse, C. J., & Waters, A. P. (2006). Regulation

- of sexual development of Plasmodium by translational repression. *Science (New York, N.Y.)*, 313(5787), 667–669.
- Manzoni, G., Marinach, C., Topçu, S., Briquet, S., Grand, M., Tolle, M., Gransagne, M., Lescar, J., Andolina, C., Franetich, J.-F., Zeisel, M. B., Huby, T., Rubinstein, E., Snounou, G., Mazier, D., Nosten, F., Baumert, T. F., & Silvie, O. (2017). Plasmodium P36 determines host cell receptor usage during sporozoite invasion. *ELife*, 6. <https://doi.org/10.7554/eLife.25903>
- Matz, J. M., & Matuschewski, K. (2018). An in silico down-scaling approach uncovers novel constituents of the Plasmodium-containing vacuole. *Scientific Reports*, 8(1), 14055.
- Pastrana-Mena, R., Dinglasan, R. R., Franke-Fayard, B., Vega-Rodríguez, J., Fuentes-Caraballo, M., Baerga-Ortiz, A., Coppens, I., Jacobs-Lorena, M., Janse, C. J., & Serrano, A. E. (2010). Glutathione reductase-null malaria parasites have normal blood stage growth but arrest during development in the mosquito. *The Journal of Biological Chemistry*, 285(35), 27045–27056.
- Rijpma, S. R., van der Velden, M., Annoura, T., Matz, J. M., Kenthirapalan, S., Kooij, T. W. A., Matuschewski, K., van Gemert, G.-J., van de Vegte-Bolmer, M., Siebelink-Stoter, R., Graumans, W., Ramesar, J., Klop, O., Russel, F. G. M., Sauerwein, R. W., Janse, C. J., Franke-Fayard, B. M., & Koenderink, J. B. (2016). Vital and dispensable roles of Plasmodium multidrug resistance transporters during blood- and mosquito-stage development. *Molecular Microbiology*, 101(1), 78–91.
- Sayers, C., Pandey, V., Balakrishnan, A., Michie, K., Svedberg, D., Hunziker, M., Pardo, M., Choudhary, J., Berntsson, R., & Billker, O. (2024). Systematic screens for fertility genes essential for malaria parasite transmission reveal conserved aspects of sex in a divergent eukaryote. *Cell Systems*, 15(11), 1075-1091.e6.
- Srivastava, A., Philip, N., Hughes, K. R., Georgiou, K., MacRae, J. I., Barrett, M. P., Creek, D. J., McConville, M. J., & Waters, A. P. (2016). Stage-specific changes in Plasmodium metabolism required for differentiation and adaptation to different host and vector environments. *PLoS Pathogens*, 12(12), e1006094.

Reviewer #1 (Remarks to the Author)

In this revised version of the manuscript, the authors have addressed some of the comments.

I have three remaining points.

1> In Supplementary figure 3C, there are two main bands for CTRL-3HA protein in the immunoblot. One is the product with expected size, what is the other band? Unspecific protein or degraded product? It is better to describe them.

2> Throughout the figures, it is good practice to provide the experiment repeat information for each experiment in the legend. One, two, or three independent experiment performed with similar results?

3> Provide the raw image for all the immunoblot.(Remarks on code availability)

Author response: In response to reviewer 1 we now describe the smaller band in Supplementary Figure 1C (not 3C) as a degradation product since it is specific to the CTRL-3HA line. Numbers of replicates are now shown in the figure legends also for the supplemental figures. The manuscript already shows full versions of all immunoblots (Fig. S1C and Fig. S5C), with the exception of the tubulin loading control in S5C. We are embarrassed to report that we did not retain the full scan of the loading control. Tubulin is a routine control and the rest of the blot showed nothing unexpected, but we recognise that we should have retained the image.

Reviewer #2 (Remarks to the Author)

This is an excellent revision by Oliver Billker and coauthors. They provide a comprehensive rebuttal to the detailed reviews. My comments as Reviewer #2 were addressed well. I am fully satisfied with their revised manuscript and find this to be an excellent study for Nature Communications.

Minor:

Lines 205-7: To obtain accurate homing rates mosquitoes each mosquito included in the analysis... Can delete "mosquitoes" or state "in mosquitoes"

Also, can the authors specify whether AlphaFold predicts 10 transmembrane domains in CTRL. Are they predicted to be aligned mostly as antiparallel pairs as in PfCRT? It would seem so from Figure 6 but would be helpful to spell this out in the text.

(Remarks on code availability)

Author response: In response to reviewer 2, we improved the language as suggested. We can also confirm that the AF2-predicted structure of CTRL consists of ten transmembrane helices arranged in antiparallel pairs and now point this out in the text.

Reviewer #4 (Remarks to the Author), on behalf of reviewer #3

The revised manuscript has adequately addressed the major and minor concerns raised in the initial review. The study presents a novel CRISPR homing screen to investigate Plasmodium oocyst-stage gene functions, an area of significant interest for malaria transmission research. The discovery of CTRL as a critical component of the oocyst digestive vacuole is a significant contribution.

The expanded discussion on gene selection rationale, homing efficiency differences between male and female gametes, and potential biases in pooled screens strengthens the study's impact. The integration of transcriptomic data and refined figures further enhance clarity. (Remarks on code availability)